



# 1 Uncertainty assessment of a dominant-process catchment

# 2 model of dissolved phosphorus transfer

**R. Dupas[1], J. Salmon-Monviola[1], K. Beven[2], P. Durand[1], P.M. Haygarth[2], M.J.**
**Hollaway[2], C. Gascuel-Odoux[1]**
[1] INRA, Agrocampus Ouest, UMR1069 SAS, F-35000 Rennes, France
[2] Lancaster Environment Centre, Lancaster University, Lancaster, United Kingdom, LA1
4YQ
Correspondence to: R. Dupas (remi.dpas@gmail.com)

## 11 Abstract

We developed a parsimonious topography-based hydrologic model coupled with a soil
biogeochemistry sub-model in order to improve understanding and prediction of Soluble
Reactive Phosphorus (SRP) transfer in agricultural headwater catchments. The model
structure aims to capture the dominant hydrological and biogeochemical processes identified
from multiscale observations in a research catchment (Kervidy-Naizin, 5 km²). Groundwater
fluctuations, responsible for the connection of soil SRP production zones to the stream, were
simulated with a fully-distributed hydrologic model at 20 m resolution. The spatial variability
of the soil phosphorus status and the temporal variability of soil moisture and temperature,
which had previously been identified as key controlling factor of SRP solubilisation in soils,
were included as part of an empirical soil biogeochemistry sub-model. The modelling
approach included an analysis of the information contained in the calibration data and
propagation of uncertainty in model predictions using a GLUE "limits of acceptability"
framework. Overall, the model appeared to perform well given the uncertainty in the
observational data, with a Nash-Sutcliffe efficiency on daily SRP loads between 0.1 and 0.8
for acceptable models. The role of hydrological connectivity via groundwater fluctuation, and
the role of increased SRP solubilisation following dry/hot periods were captured well. We
conclude that in the absence of near continuous monitoring, the amount of information
contained in the data is limited hence parsimonious models are more relevant than highly




parameterised models. An analysis of uncertainty in the data is recommended for model
calibration in order to provide reliable predictions.
**1   Introduction**
Excessive phosphorus (P) concentrations in freshwater bodies result in increased
eutrophication risk worldwide (Carpenter et al., 1998; Schindler et al., 2008). Eutrophication
restricts economic use of water and poses a serious health hazard to humans, due to the
potential development of harmful cyanobacteria (Bradley et al., 2013; Serrano et al., 2015). In
western countries, reduction of point source P emissions in the last two decades has resulted
in a proportionally increasing contribution of diffuse sources, mainly from agricultural origin
(Alexander et al., 2008; Grizzetti et al., 2012; Dupas et al., 2015a). Of particular concern are
dissolved P forms, often measured as Soluble Reactive Phosphorus (SRP), because they are
highly bioavailable and therefore a likely contributor to eutrophication.
To reduce SRP transfer from agricultural soils it is important to identify the spatial origin of P
sources in agricultural landscapes, the biogeochemical mechanisms causing SRP
solubilisation in soils and the dominant transfer pathways. Research catchments provide
useful data to investigate SRP transport mechanisms: typically, the temporal variations in
water quality parameters at the outlet, together with hydroclimatic variables, are investigated
to infer spatial origin and dominant transfer pathways of SRP (Haygarth et al., 2012; Outram
et al., 2014; Dupas et al., 2015b; Mellander et al., 2015; Perks et al., 2015). Hypotheses
drawn from analysis of water quality time series can be further investigated through hillslope
monitoring and/or laboratory experiments (Heathwaite and Dils, 2000; Siwek et al., 2013;
Dupas et al., 2015c). When dominant processes are considered reasonably known, it is
possible to develop computer models, for two main purposes: first, to validate scientific
conceptual models, by testing whether model predictions can produce reasonable simulations
compared to observations. Of particular interest is the possibility to test the capability of a
computer model to upscale P processes observed at fine spatial resolution (soil column,
hillslope) to a whole catchment. Second, if the models survive such validation tests, then they
can be useful tools to simulate the response of a catchment system to a future perturbation
such as changes in agricultural management and climate changes.
However, process-based P models generally perform poorly compared to, for example,
nitrogen models (Wade et al., 2002; Dean et al., 2009; Jackson-Blake et al., 2015a). This is of
major concern because poor model performance suggests poor knowledge of dominant





processes at the catchment scale, and poor reliability of the modelling tools used to support
management. The origin of poor model performance might be conceptual misrepresentations,
structural imperfection, calibration problems, irrelevant model evaluation criteria and
difficulties in properly assessing the information content of the available data when it is
subject to epistemic error. All five causes of poor model performance are intertwined, e.g.
model calibration strategy depends on model performance evaluation criteria, which depend
on the way the information contained in the observation data is assessed (Beven and Smith,

8  2015).

A key issue in environmental modelling is the level of complexity one should seek to
incorporate in a model structure. Several existing P transfer models, such as INCA (Wade et
al., 2002), SWAT (Arnold et al., 1998) and HYPE (Lindstrom et al., 2010) seek to simulate
many processes, with the view that complex models are necessary to understand processes
and to predict the likely consequences of land-use or climate changes. However, these
complex models include many parameters that need to be calibrated, while the amount of data
available for calibration is often low. An imbalance between calibration requirement and the
amount of available observation data can lead to equifinality issues, i.e. when many model
structures or parameter sets lead to acceptable simulation results (Beven, 2006). A
consequence of equifinality is the risk of unreliable prediction when an "optimal" set of
parameters is used (Kirchner, 2006), and large uncertainty intervals when Monte Carlo
simulations are performed (Dean et al., 2009).  In this situation, it will be worth exploring
parsimonious models that aim to capture the dominant hydrological and biogeochemical
processes controlling SRP transfer in agricultural catchment. For example, Hahn et al. (2013)
used a soil-type based rainfall-runoff model (Lazzarotto et al., 2006) combined with an
empirical model of soil SRP release derived from rainfall simulation experiments over soils
with different P content and manure application level/timing (Hahn et al., 2012) to simulate
daily SRP load from critical sources areas.
A second key issue, linked to the question of model complexity, concerns model calibration
and evaluation. Both calibration and evaluation require assessing the fit of model outputs with
observation data. However, observation data are generally not directly comparable with model
outputs, because of incommensurability issues and/or because they contain errors (Beven,
2006; 2009). Typically, predicted daily concentrations and/or loads are evaluated against data
from grab samples collected on a daily or weekly basis. The information content of these data


must be carefully evaluated to propagate uncertainty in the data into model predictions.
Uncertainty in grab sample data might stem from i) sampling frequency problems and ii)
measurement problems (Lloyd et al., 2015). Grab sample data represent a snapshot of the
concentration at a given time of the day, which can differ from the flow weighted daily
concentration. This difference between observation data and simulation output can be large
during storm events in small agricultural catchments, as P concentrations can vary by several
orders of magnitudes during the same day (Heathwaite and Dils, 2000; Sharpley et al., 2008).
Model evaluation can be severely penalised by this difference, because many popular
evaluation criteria such as the Nash-Sutcliffe efficiency (NSE) are sensitive to extreme values
and errors in timing (Moriasi et al.,2007). During baseflow periods, it is more likely that grab
sample data are comparable to flow-weighted mean daily concentrations, as concentrations
vary little during the day and they are usually low in the absence of point sources. However,
measurement errors are expected to occur at low concentrations, either due to too long storage
times or laboratory imprecision when concentrations come close to detection/quantification
limits (Jarvie et al., 2002; Moore and Locke, 2013). Uncertainty in the data can also relate to
discharge measurement and input data (e.g. maps of soil P content and rainfall data). In this
paper we strive to identify and quantify the different sources of uncertainty in the data when
the required quality check tests have been performed. A Generalised Likelihood Uncertainty
Estimation (GLUE) "limits of acceptability" approach (Beven, 2006; Beven and Smith, 2015)
is used to calibrate/evaluate the model.
This paper presents a dominant-process model that couples a topography-based hydrologic
model with a soil biogeochemistry sub-model able to simulate daily discharge and SRP loads.
The dominant processes included in the hydrologic and soil biogeochemistry sub-models have
been identified in previous analyses of multiscale observational data, which have
demonstrated on the one hand the control of groundwater fluctuation on connecting soil SRP
production zones to the stream (Haygarth et al., 2012; Jordan et al., 2012; Dupas et al., 2015b;
2015d; Mellander et al., 2015), and on the other hand the role of antecedent soil moisture and
temperature conditions on SRP solubilisation in soils (Turner and Haygarth, 2001; Blackwell
et al., 2009; Dupas et al., 2015c). Model development and application was performed in the
Kervidy-Naizin catchment in western France with the objectives of: i) testing if the model
was capable of capturing daily variation of SRP load, thus confirming hypotheses on
dominant processes; ii) develop a methodology to analyse and propagate uncertainty in the




data into model prediction using a "limits of acceptability" approach. Model development and
analysis of uncertainty in the data are interlinked in this approach.

## 3   2   Material and methods

### 4   2.1   Study catchment

#### 5   2.1.1   Site description

Kervidy–Naizin is a small (4.94 km²) agricultural catchment located in central Brittany,
Western France (48°N, 3°W). It belongs to the AgrHyS environmental research observatory
(http://www6.inra.fr/ore_agrhys_eng), which studies the impact of agricultural activities and
climate change on water quality (Molenat et al., 2008; Aubert et al., 2013; Salmon-Monviola
et al., 2013; Humbert et al., 2014). The catchment (Fig. 1) is drained by a stream of second
Strahler order, which generally dries up in August and September. The climate is temperate
oceanic, with annual cumulative precipitation and specific discharge averaging $854 \pm 179$ mm
and $290 \pm 106$ mm, respectively, from 2000 to 2014. Mean annual temperature is $11.2 \pm$
$0.6$°C. Elevation ranges from 93 to 135 m above sea level. Topography is gentle, with
maximum slopes not exceeding 5%. The bedrock consists of impervious, locally fractured
Brioverian schists and is capped by several metres of unconsolidated weathered material and
silty, loamy soils. The hydrological behaviour is dominated by the development of a water
table that varies seasonally along the hillslope. In the upland domain, consisting of well
drained soils, the water table remains below the soil surface throughout the year, varying in
depth from 1 to > 8 m. In the wetland domain, developed near the stream and consisting of
hydromorphic soils, the water table is shallower, remaining near the soil surface generally
from October to April each year. The land use is mostly agriculture, specifically arable crops
and confined animal production (dairy cows and pigs). A farm survey conducted in 2013 led
to the following land use subdivisions: 35% cereal crops, 36% maize, 16% grassland and 13%
other crops (rape seed, vegetables). Animal density was estimated as high as 13 livestock
units ha$^{-1}$ in 2010. Estimated soil P surplus is 13.1 kg P ha$^{-1}$ yr$^{-1}$ (Dupas et al., 2015b) and soil
extractable P in 2013 (Olsen et al., 1954) is $59 \pm 31$ mg P kg$^{-1}$ (n = 89 samples). A survey
targeting riparian areas highlighted the legacy of high soil P content in these currently
unfertilized areas (Dupas et al., 2015c). No point source emissions are recorded but scattered
dwellings with septic tanks are present in the catchment.



### 2.1.2 Hydroclimatic and chemical monitoring

Kervidy-Naizin was equipped with a weather station (Cimel Enerco 516i) located 1.1 km from the catchment outlet. It recorded hourly precipitation, air and soil temperatures, air humidity, global radiation, wind direction and speed, and estimates Penman evapotranspiration. Stream discharge was estimated at the outlet with a rating curve and stage measurements from a float-operator sensor (Thalimèdes OTT) upstream of a rectangular weir.

To record both seasonal and within storm dynamics in P concentration, two monitoring strategies complemented each other from October 2013 to August 2015: a daily manual grab sampling at approximately the same time (between 16:00 – 18:00 local time) and automatic high frequency sampling during 14 storm events (autosampler ISCO 6712 Full-Size Portable Sampler, 24 one litre bottles filled every 30 min). The water samples were filtered on-site, immediately after grab sampling and after 1-2 days in the case of autosampling. They were analysed for SRP (ISO 15681) within a fortnight. To assess uncertainty in daily SRP concentration related to sampling time, storage and measurement errors, a second grab sample was taken at a different time of the day (between 11:00 – 15:00 local time) in 36 instances during the study period. The second sample was analysed within 24h with the same method; this second dataset is referred to as verification dataset, as opposed to the reference dataset. Among the 36 pairs of comparable daily samples, 12 were taken during storm events and 24 during baseflow periods. To assess uncertainty in high frequency SRP concentration during storm events due to delayed filtration of autosampler bottles, 5 grab samples were taken during the course of 4 distinct storms and were filtered immediately. The same lab procedure was used to analyse SRP.

### 2.1.3 Identification of dominant processes from multiscale observations

Observations in the Kervidy-Naizin catchment have highlighted that the temporal variability in stream SRP concentrations could not be related to the calendar of agricultural practices, but rather to hydrological and biogeochemical processes (Dupas et al., 2015b). The primary control of hydrology on SRP transfer has also been evidenced in several other small agricultural catchments (e.g. Haygarth et al, 2012; Jordan et al., 2012; Mellander et al., 2015). In the Kervidy-Naizin catchment, groundwater fluctuations in valley bottom areas was identified as the main driving factor of SRP transfer, through the hydrological connectivity it creates when it intercepts shallow soil layers (Dupas et al., 2015b).



In-situ monitoring of soil pore water at 4 sites (15 cm and 50 cm depths) in the Kervidy-
Naizin catchment has shown that mean SRP concentration in soils was a linear function of
Olsen P (Olsen et al., 1954). This reflects current knowledge that a soil P test, or alternatively
estimation of a degree of P saturation, can be used to assess solubilisation in soils
(Beauchemin and Simard, 1999; McDowell et al., 2002; Schoumans et al., 2015). This linear
relationship derived from the data contrasts however with other studies, where threshold
values above which SRP solubilisation increases greatly have been identified (Heckrath et al.,
1995; Maguire et al., 2002).
Soluble Reactive Phosphorus solubilisation in soil varies seasonally according to antecedent
conditions of temperature and soil moisture. Dry and/or hot conditions are favourable to
accumulation of mobile P forms in soils, while water saturated conditions lead to their
flushing (Turner et al., 2001; Blackwell et al., 2009; Dupas et al., 2015c).

## 13  2.2 Description of the Topography-based Nutrient Transfer and
## 14        Transformation – Phosphorus (TNT2-P)

TNT2 was originally developed as a process-based and spatially explicit model simulating
water and nitrogen fluxes at a daily time step (Beaujouan et al., 2002) in meso-scale
catchments ($< 50$ km$^2$). TNT2-N has been widely used for operational objectives, to test the
effect of mitigation options proposed by local stakeholders or public policy-makers (Moreau
et al., 2012; Durand et al., 2015), on nitrate fluxes and concentrations in rivers.
TNT2-P uses a modified version of the hydrological sub-model in TNT2-N, to which a
biogeochemistry sub-model was added to simulate SRP solubilisation in soils.

### 22  2.2.1 Hydrological sub-model

The assumptions in the hydrological sub-model are derived from TOPMODEL which has
previously been applied to the Naizin catchment (Bruneau et al., 1995; Franks et al., 1998): 1)
the effective hydraulic gradient of the saturated zone is approximated by the local topographic
surface gradient (tan β). It is calculated in each cell of a Digital Elevation Model (DEM) at the
beginning of the simulation; 2) the effective downslope transmissivity (parameter T) of the
soil profile in each cell of the DEM is a function of the soil moisture deficit (Sd). Hydraulic
conductivity decreases exponentially with depth (parameter m, Fig. 2). Hence water fluxes (q)
are computed as:





$$q = T * tan\beta * \exp(-\frac{Sd}{m})$$  (1)
Based on these assumptions, TNT2 computes an explicit cell-to-cell routing of fluxes, using a
D8 algorithm. This explicit cell-to-cell routing of fluxes increases computation times
compared to TOPMODEL, for which calculations are grouped according to a distribution of
hydrologically similar points, but it allows taking account of spatial interactions between soil
and groundwater, which has been shown to improve representation of nutrients fluxes and
transformations (Beaujouan et al., 2002).
To simulate SRP fluxes, the only modification to the hydrological sub-model aimed to
compute water fluxes from each soil layer by integrating [1] between the maximum depth of
the soil layer considered and:
- estimated groundwater level, if the groundwater table is within the soil layer
considered
or
- the minimum depth of the soil layer considered, if the groundwater table above the
soil layer considered
In this application of the TNT2-P model, 5 soil layers with a thickness of 10 cm are
considered. Hence, 7 flow components are computed in the model:
- overland flow on saturated surface
- 5 sub-surface flow components, for each soil layer
- deep flow, i.e. flow below the 5 soil layers
**2.2.2  Soil-P sub-model**
The soil-P sub-model is empirically derived from soil pore water monitoring data (Dupas et
al., 2015c), specifically assuming that:
- background SRP concentration in the soil pore water of a given layer is proportional to
soil Olsen P;
- seasonal increases in P availability compared to background conditions are determined
by biogeochemical processes, controlled by antecedent temperature and soil moisture.
Data show that SRP availability in the soil pore water increases following periods of
dry and hot conditions (Dupas et al., 2015c).





Hence, SRP transfer is modelled with parameters that describe both mobilisation and transfer
to the stream. A different parameter is used to simulate transfer via overland flow and sub-
surface flow.
$F_{SRP\ overland} = Coef_{SRP\ overland} * P_{Olsen} * q_{overland}$ (2)
$F_{SRP\ sub-surface} = Coef_{SRP\ sub-surface} * P_{Olsen} * q_{sub-surface}$ (3)
Where $F_{SRP\ overland}$ and $F_{SRP\ sub-surface}$ are SRP transfer via overland flow and sub-surface
flow for a given soil layer respectively, $q_{overland}$ and $q_{sub-surface}$ are water flows from the
same pathways. $Coef_{SRP\ overland}$ and $Coef_{SRP\ sub-surface}$ are coefficients which vary
according to antecedent temperature and soil moisture conditions, such as:
$Coef_{SRP} = Coef_{background} * (1 + F_T * F_S)$ (4)
Where $Coef_{SRP}$ is either $Coef_{SRP\ overland}$ or $Coef_{SRP\ sub-surface}$, and $F_T$ and $F_S$ are
temperature and soil moisture factors, respectively. $F_T$ and $F_S$ are expressed as:
$F_T = \exp(\frac{mean(temperature, i\ days) - T1}{T2})$ (5)
$F_S = 1 - \left(\frac{mean(water\ concentent, i\ days)}{maximum\ water\ content}\right)^{S1}$ (6)
Where T1, T2 and S1 are calibrated coefficients. The antecedent condition time length
consists in a period of i=100 days. Both soil temperature and soil moisture are estimated by
TNT2 soil module (Moreau et al., 2013). Because soil moisture in the deep soil layers can
differ significantly from that of shallow soil layers, two values of $F_S$ are calculated for two
soil depth 0-20 cm and 20-50 cm. The temperature factor $F_T$ was calculated as an average
value for the entire soil profile 0-50 cm. Contrary to water fluxes, SRP fluxes are not routed
cell-to-cell, because we lacked knowledge of the rate of SRP re-adsorption in downslope
cells, and on the long term fate of re-adsorbed SRP. Hence, all the SRP emitted from each cell
through overland flow and sub-surface flow reaches the stream on the same day. For deep
flow, only the immediate riparian flux is used in determining SRP inputs to the river.
**2.2.3 Input data and parameters**
Spatial input data include:
- A DEM in raster format. Here, a 20 m resolution DEM was used, hence model
calculations were made in 12348 grid cells covering a 4.94 km$^2$ catchment.





1    -    A map of soils with homogeneous hydrological parameter value, in raster format.

2         Here, two soil classes were considered by differentiating well-drained (86%) and

3         poorly drained soils (14%) according to Curmi et al. (1998) (Fig. 1).

4    -    A map of surface Olsen P in raster format and description of decrease in P Olsen with

5         depth for five soil layers between 0-50 cm. Here, the map of Olsen P in the 0-15 cm

6         soil layer was obtained from statistical modelling with the rule-based regression

7         algorithm CUBIST (Quinlan, 1992) using data from 198 soil samples (2013) in an

8         area of 12 km² encompassing the 4.94 km² catchment (Matos-Moreira et al., 2015).

9         To describe how P Olsen decreases with depth, land use information was used. In

10        tilled fields, i.e. all crop rotations including arable crops, Olsen P was assumed to be

11        constant between 0-30 cm and to decrease linearly with depth between 30-50 cm. In

12        no-till fields, i.e. permanent pasture and woodland, Olsen P was assumed to decrease

13        linearly with depth between 0-50 cm. An exponential decrease with depth is more

14        commonly adopted in untilled land (e.g. Haygarth et al., 1998; Page et al., 2005), but a

15        specific sampling in currently untilled areas in the Kervidy-Naizin catchment (Dupas

16        et al., 2015c) has shown that a linear function is more appropriate, probably because

17        of these areas having been ploughed in the past.

Climate input data include minimum and maximum air temperature, precipitation, potential
evapotranspiration, global radiation on a daily basis. The TNT2 model allows for several
climate zones to be considered, in which case a raster map of climate zone must be provided
to the model. Here, only one climate zone is considered.
In total, the TNT2-P model includes 15 parameters for each soil type, i.e. 30 parameters in
total if two soil drainage classes are considered. To reduce the number of model runs
necessary to explore the parameter space using Monte Carlo simulations, several parameters
were given fixed values, or a constant ratio between the two soil types was set (Table 1). In
the hydrological sub-model, the parameters to vary were identified in a previous sensitivity
analysis (Moreau et al., 2013). In the soil sub-model, all the parameters were varied.
Finally, only 12 parameters were varied independently. Initial parameter ranges for the
hydrological sub-model were based on literature-derived values (Moreau et al., 2013) and
those for the soil sub-model were based on a preliminary manual trial and error procedure.
The SRP concentration for deep flow water was based on actual measurement of SRP in the





weathered schist (Dupas et al., 2015c). A constant flux value for domestic sources was set at
the 1% percentile of the daily flux between 2007 and 2013 (Dupas et al., 2015b).
**2.3    Deriving limits of acceptability from data uncertainty assessment**
The Monte Carlo based Generalized Likelihood Uncertainty Estimation (GLUE)
methodology has been widely used in hydrology and is described elsewhere (Beven and
Freer, 2001; Beven, 2006, 2009). Briefly, the rationale of GLUE is that many model
structures and parameter sets can give "acceptable" results, according to one or several
performance measures, due to equifinality. Hence, GLUE considers that all models that give
acceptable results should be used for prediction. A key issue in GLUE is to decide on a
performance threshold to define acceptable models; typically, modellers set a threshold value
of a measure such as the Nash-Sutcliffe Efficiency based on their subjective appreciation of
data uncertainty or on previously used values. To allow for a more explicit justification of the
performance threshold values used, the limits of acceptability approach outlined by Beven
(2006) relies on an assessment of uncertainty in the calibration/evaluation data. According to
this approach, all model realisations that fall within the limits of acceptability are used for
prediction, weighted by a score calculated based on overall performance.
Details on how the limits of acceptability for daily discharge and daily SRP load were derived
from uncertainty assessment of the observational data are presented below. Input data, such as
weather and soil data, also contained uncertainty which were not accounted for in the limits of
acceptability due to a lack of data to quantifying them.
**2.3.1    Discharge**
Error in discharge measurement data was assessed from the original discharge measurements
used to calibrate the stage-discharge rating curve (Carluer, 1998). The rating curve used in
this study was:
$Q = a * (h - h_0)^b$                                   (7)
Where Q is discharge, h is stage reading, $h_0$ is stage reading at zero discharge, a and b are
calibrated coefficients. Limits of acceptability were defined as the 90% prediction interval of
log-log linear regression (Fig. 3). Estimated acceptability range was ±39% on average. For
daily discharge values below 2 mm d$^{-1}$, fixed acceptability limits were set at the 90%
prediction interval for a stage measurement corresponding to 2 mm d$^{-1}$.





### 2.3.2  SRP load

Uncertainty in "observed" daily load includes uncertainty in discharge (see 2.3.1.) and uncertainty in SRP concentration. Uncertainty in daily load was estimated summing up relative uncertainty assessed for discharge and SRP concentration. Uncertainty in SRP concentration stems from sampling frequency problems as one grab sample collected on a specific day is incommensurable with the mean daily concentration or load simulated by the model. Further, measurement errors exist that include the effect of storage time (Haygarth et al., 1995). During baseflow periods, measurement error was expected to be the main source of uncertainty because relative measurement error is large for low concentrations, especially when sample storage time exceeds 48h (Jarvie et al., 2002), while concentrations vary little. During storm events, sampling frequency was expected to be the main source of uncertainty because SRP concentration can vary by one order of magnitude within a few hours. Therefore, different acceptability limits were set for both flow conditions. We considered storms as events with $> 20 \, l \, s^{-1}$ increase in discharge and the following 24h.

During baseflow periods, the acceptability limits were derived from the 90% prediction interval of a linear regression model linking pairs of data points sampled on the same day (reference sample between 16:00-18:00, verification sample between 11:00-15:00) and analysed independently (within a fortnight for the reference sample and within 1-2 days for the verification sample). It was assumed that there was no systematic bias between the two datasets due to different sampling time. The reference SRP concentrations were on average 13% lower than the verification value but this difference was not statistically significant (Mann-Whitney Rank Sum Test, $p > 0.05$). Hence, the expected underestimation of SRP concentration due to long sample storage appears to be overshadowed by other sources of uncertainty such as variability in SRP concentration during the day of sampling or analytical imprecision at low concentrations. This method encompasses all various sources of uncertainty, which results in prediction intervals much wider than what would result from a mere repeatability test: at the median concentration ($0.02 \, mg \, l^{-1}$), estimated prediction interval was 166% with this method versus 57% with a repeatability test (Fig. 4).

During storm events, acceptability limits were derived from the 90% prediction interval of concentration discharge empirical models $C= a*Q^b$ using high frequency autosampler data. A distinct empirical model was used to fit to each storm event monitored and a delay term was introduced manually in the empirical model when a time lag existed between



concentration and discharge peaks. The empirical models were then applied to extrapolate
concentration estimation during two days at 10 min resolution, for each of the 14 storm events
monitored. Finally the 2-day mean "observed" load was estimated as the mean of 10 min
loads and uncertainty limits were derived from the 90% prediction interval. In model
evaluation, the mean of simulated loads during 2 consecutive days was evaluated against the
2-day mean "observed" load for which prediction intervals have been calculated. A 2-day
acceptability limit enables to cover the whole of storm events (Fig. 5 and Supplement).
When comparing autosampler data with data from immediately filtered samples, the ratio
obtained ranged 1-1.6 (mean = 1.3), hence autosampler data were underestimated arguably
through adsorption or biological consumption. We used the mean ratio to correct all storm
uncertainty intervals by 30% and the range values to extend the upper limit by 60%. During
days with a storm event not monitored at high frequency with an autosampler, we considered
that the grab sample data did not contain enough information to derive an acceptability
interval for daily SRP load.

### 15  2.3.3  Model runs and selection of acceptable models

To explore the parameter space, 15,000 Monte Carlo realisations were performed to simulate
daily discharge and SRP load during the water years 2013-2014 and 2014-2015. A 7-month
initialisation period was run to reduce the impact of initial conditions on simulated results
during the study period, from 1 October 2013 to 31 July 2015.
To be considered acceptable, model runs must fall within the acceptability limits defined in
2.3.1 and 2.3.2. More specifically, 100% of simulated daily discharge, 100% of simulated
baseflow SRP load and 100% of simulated storm SRP load had to fall within the acceptability
limits. Thus, 572 acceptability tests were performed for discharge, 378 for baseflow SRP load
and 14 for storm SRP loads, i.e. 964 evaluation criteria.
To evaluate the model performance in more detail, normalized scores were calculated during
6 periods (Table 2). To calculate the scores, a difference was calculated between each of the
daily simulated discharge, baseflow SRP load and 2-day storm SRP loads and the
corresponding observation. This difference was then normalized by the width of the
acceptability limit defined for that day, so the score has a value of 0 in the case of a perfect
match with observation, -1 at the lower limit and +1 at the upper limit (Fig. 6a). Finally, the
median of this ratio was calculated for each of the 6 periods to investigate whether the model





tended to underestimate or overestimate discharge and loads at different moments of the year
and between the two years.
Model runs were successively evaluated for discharge, baseflow SRP load and storm SRP
load. To use the models for prediction, each accepted model was given a likelihood weight
according to how well it has performed for each of the 964 evaluation criteria. Here a
triangular weight was calculated for each evaluation criteria (Fig. 5 b), with the base of the
triangle corresponding to the acceptability limit. Calculated weights were then averaged for
discharge, baseflow SRP load and storm SRP load respectively and the final likelihood was
calculated as the sum of all three averages.
The model's sensitivity to each hydrological and soil parameter was performed with a
Hornberger-Spear-Young Generalised Sensitivity Analysis (HSY GSA, Whitehead and
Young, 1979; Hornberger and Spear, 1981). For each evaluation criteria (daily discharge,
daily baseflow SRP load, 2-day storm SRP load), the model runs were split into acceptable
and non-acceptable runs according to the above-mentioned acceptability limits.   Then a
Kolmogorov-Smirnov test is performed to assess whether the distribution of each of the three
evaluation criteria differ between acceptable and non-acceptable models for each parameter.
The p value of the Kolmogorov-Smirnov test is used to discriminate whether the model is
critically sensitive ($p<0.01$ '***'), importantly sensitive ($p<0.1$ '*') or insignificantly
sensitive ($p>0.1$ '.') to each parameter and for each of the three evaluation criteria. Because
the Kolmogorov-Smirnov test might suggest that small differences in distribution are very
significant when there are larger number of runs, this method is a qualitative guide to relative
sensitivity.
In addition to acceptability limit approach, a NSE (Moriasi et al., 2007) was calculated for
daily discharge and daily load and concentration to allow comparison with other modelling
studies where is has been taken as an evaluation criteria.
**3   Results**
**3.1   Presentation of observation data and calculation of acceptability limits**
The two water years studied were highly contrasted in terms of hydrology and SRP loads.
Water year 2013-2014 was the wettest in the last 10 years, with cumulative rainfall 1289 mm
and cumulative runoff 716 mm. Water year 2014-2015 was an average year (5[th] wettest in the
last 10 years), with cumulative rainfall 677 mm and cumulative runoff 383 mm. Annual SRP





load was 0.35 kg P ha$^{-1}$ yr$^{-1}$ in 2013-2014 and 0.17 kg P ha$^{-1}$ yr$^{-1}$ in 2014-2015, i.e. a
difference 10% higher than that of discharge. Observed mean SRP concentration during the
study period was 0.024 mg l$^{-1}$.
Fig. 7 shows acceptability limits for daily discharge and daily SRP loads. Note that
acceptability limits for discharge were calculated every day, while acceptability limits for
SRP load was calculated on a daily basis during baseflow periods and on a 2-day basis during
storm events monitored at high frequency. No SRP load acceptability limit was calculated
during storm events when no high frequency autosampler data was available.
**3.2   Model evaluation**
First, model runs were evaluated against acceptability limits defined for discharge (Fig. 8).
4,120/15,000 models fulfilled the selection criterion for discharge, i.e. they had 100% of
simulated daily discharge within the acceptability limits. The NSE estimated for these models
ranged from 0.78 to 0.92. The normalized scores calculated seasonally (Fig. 9a) show that
simulated discharge is often overestimated in autumn and spring, and underestimated in
winter.
Then, model runs were evaluated against acceptability limits defined for SRP loads (Fig. 8b).
During baseflow periods, 3,730/15,000 models fulfilled the selection criterion for SRP loads,
i.e. they had 100% of simulated daily SRP load within the acceptability limits. Among them,
1,210also fulfilled the previous selection criterion for discharge. Normalized scores for
baseflow SRP load showed the same trend as for discharge (Fig. 9b), i.e. overestimation in
autumn and spring, and underestimation in winter. During storm events, only 5 models
fulfilled the selection criterion for SRP loads, i.e. they had 14/14 of simulated 2-day storm
SRP loads within the acceptability limits, but none of them also fulfilled the selection criteria
for discharge and baseflow SRP loads. Two storm events were particularly difficult to
simulate (number 2 and number 9, Fig. 9c), probably because their acceptability interval was
very narrow as a result of only small changes in discharge and concentration. To obtain a
reasonable number of acceptable models, we relaxed the selection criterion so that the
acceptable models had to simulate 12/14 of storm loads within the acceptability limits, in
addition to the selection criteria defined for discharge and baseflow SRP load: 418 models
were then accepted. Estimated NSE of these 418 models ranged from 0.09 to 0.80 for daily





load and from negative values to 0.53 for daily concentrations (this includes all data from the
regular sampling).

### 3.3   Sensitivity analysis and prediction results

According to the HSA generalised sensitivity analysis, simulated discharge was critically
sensitive to 10 out of the 12 hydrological parameters varied. Simulated SRP load was
critically sensitive to the sub-surface and overland flow parameters during baseflow periods
and to the overland flow parameter during storm events. During baseflow periods, SRP load
was insignificantly sensitive to the parameter associated with deep flow load. Both baseflow
and storm SRP loads were critically sensitive to the parameter related to soil moisture and soil
temperature dependent SRP solubilisation (S1, T1 and T2), in addition to respectively 11 and
8 hydrological parameters.
Fig. 10 shows the daily discharge, SRP load and concentration as simulated by the acceptable
models. Simulated SRP load during the water year 2013-2014 ranged $0.77 - 3.28$ kg P ha$^{-1}$ yr$^{-1}$
(median = 1.62 kg P ha$^{-1}$ yr$^{-1}$); simulated SRP load during the water year 2014-2015 ranged
$0.14 - 0.73$ kg P ha$^{-1}$ yr$^{-1}$ (median = 0.32 kg P ha$^{-1}$ yr$^{-1}$). Best estimate of SRP load according
to observation data was 0.35 kg P ha$^{-1}$ yr$^{-1}$ in 2013-2014 and 0.17 kg P ha$^{-1}$ yr$^{-1}$ in 2014-2015.
According to the model, $56 - 61\%$ (median = 58%) of water discharge and $71 - 75\%$ (median
= 62%) of SRP load occurred during storm events. Mean SRP concentrations during the two
water years ranged $0.013 - 0.043$ mg l$^{-1}$ (median = 0.028 mg l$^{-1}$), while mean observed SRP
concentration was 0.024 mg l$^{-1}$.

### 4   Discussion

### 4.1   Role of hydrology and biogeochemistry in determining SRP transfer

The fairly good performance of TNT2-P at simulating SRP loads confirms that the
hydrological and biogeochemical processes included into the model are dominant controlling
factors in the Kervidy-Naizin catchment. The primary control of hydrology in controlling
connectivity between soils and streams has been highlighted by many studies analysing water
quality time series at the outlet of agricultural catchments (Haygarth et al., 2012; Jordan et al.,
2012; Dupas et al., 2015c; Mellander et al., 2015). This modelling exercise also confirmed
that SRP solubility was determined by the soil P Olsen content and could vary according to
temperature and moisture conditions. The underlying processes have not been identified





precisely in the Kervidy-Naizin catchment: independent laboratory experiments have shown
that microbial cell lysis resulting from alternating dry and water saturated periods in the soil
could be the cause of increased SRP mobility (Turner and Haygarth, 2001; Blackwell et al.,
2009). This could explain the moisture dependence of SRP solubility in the model.
Furthermore, net mineralisation of soil organic phosphorus could explain the temperature
dependence of SRP solubility in the model. These two hypotheses may explain increased SRP
solubility in soils in periods of dry and hot conditions and will be further explored by
incubation experiment with soils from the Kervidy-Naizin catchments.
**4.2 Potential improvements to the model structure according to modelling**
**purpose**
The TNT2-P model was designed to test hypotheses about dominant processes and for this
purpose, a parsimonious model structure was chosen to include only the processes which were
to be tested. This parsimonious model structure might contain some conceptual
misrepresentations due to oversimplification, and it might not include all the processes
necessary for the purpose of evaluating management scenarios. This section discusses
whether the simplifications made are acceptable in the context of different catchment types,
and to which conditions the model could be made more complex by including additional
routines for the purpose of evaluating management scenarios.
From a conceptual point of view, the lack of cell-to-cell routing of SRP fluxes might result in
erroneous results in some contexts. The fact that all the SRP emitted from each cell through
overland flow and sub-surface flow reaches the stream on the same day is acceptable for the
catchment studied because groundwater interception of shallow soil layers occurs in the
riparian zone only, hence the signal of SRP mobilisation in these soils is generally transmitted
to the stream (Dupas et al., 2015c). This simplification would not be acceptable in catchments
where soil-groundwater interactions are taking place throughout the landscape, e.g. due to
topographic depressions or poorly drained soils. In the latter type of catchment, transmission
of the SRP mobilisation signal to the stream is more complex to comprehend (Haygarth et al.,
2012), hence a more complex model structure would be required.
The reason for this simplification was that we lacked knowledge of SRP re-adsorption in
downslope cells and on the long-term fate of re-adsorbed SRP. For a more physically realistic
representation of processes, it is likely that an explicit representation of flow velocities and



pathways would be necessary, along with an explicit representation of several soil P pools.
However, such an explicit representation of processes contradicts the idea of a parsimonious
model, which was adopted here for the purpose of identifying dominant processes. In this
respect, TNT2-P is an aggregative model rather than a fully distributed model although it is
based on a fully distributed hydrological model (Beaujouan et al., 2002). The current spatial
distribution allows finer representation of soil-groundwater interactions than semi-distributed
models such as SWAT (Arnold et al., 1998), INCA-P (Wade et al., 2002) and HYPE
(Lindstrom et al., 2010) but at higher computation cost. It would be interesting to test to
which extent moving from an aggregative model with fully distributed information to a semi-
distributed model would degrade the model performance and in the same time reduce
computation cost.  This could be achieved by grouping cells according to a hydrological
similarity criterion like in the original TOPMODEL and Dynamic Topmodel (Beven and
Freer, 2001; Metcalfe et al., 2015) and do the same for similarity in soil P content.
If reducing the number of calculation units proved to reduce computation cost without
degrading quality of prediction, it would be possible to include more parameters in the model,
for example to simulate SRP re-absorption in downslope cells or include routines to simulate
the evolution of soil P content under different management scenarios (Vadas et al., 2011;
2012), and still perform a Monte-Carlo based analysis of uncertainty. The question of
coupling or not such a soil P routine with the current TNT2-P model will depend on available
data and on the length of available time series: studying the evolution of the soil P content
requires at least a decade of soil observation data (Ringeval et al., 2014) and probably a
longer period of stream data to account for the time delay for a perturbation in the catchment
to become visible in the stream (Wall et al., 2013). Thus, the two years of daily stream SRP in
the Kervidy-Naizin catchment are not enough to build a coupled soil-hydrology model with
an elaborate soil P routine. Therefore, as things stand, it is more reasonable to generate new
soil P Olsen maps with a separate model such as the APLE model (Vadas et al., 2012;
Benskin et al., 2014) or the 'soil P decline' model used by Wall et al. (2013), and use these
maps as input to TNT2-P.
Because the current model can simulate response to rainfall, soil moisture and temperature, it
could be used to test the effect of climate scenarios on SRP transfer. In Western France, and
more generally in Western Europe, the climate for the next few decades is expected to consist
of hotter, drier summers and warmer, wetter winter (Jacob et al., 2007; Macleod et al., 2012;





Salmon-Monviola et al., 2013) with increased frequency of high intensity rainfall events
(Dequé 2007). In these conditions, SRP concentrations and load will seemingly increase
compared to today's climate as a result of both an increase in SRP solubility in soil due to
higher temperature and more severe drought and an increase in transfer due to wetter winter
and more frequent high intensity rainfall events. TNT2-P could be used to confirm and
quantify the expected increase in SRP transfer from diffuse sources in future climate
conditions.
### 4.3   Improving information content in the data
Despite relatively large uncertainty in the data used in this study, it was possible to build a
parsimonious catchment model of SRP transfer for the purpose of testing hypotheses about
dominant processes, namely the role of hydrology in controlling connectivity between soils
and streams and the role of temperature and moisture conditions in controlling soil SRP
solubilisation. However, the large uncertainties in the calibration data lead to large prediction
uncertainty. For example, the SRP load estimated by the behavioural models from 2013 to
2015 ranged from 0.45 to 2.0 kg P ha$^{-1}$ yr$^{-1}$; hence the width of the credibility interval was
160% of the median (0.97 kg P ha$^{-1}$ yr$^{-1}$). Similarly, the mean SRP concentration estimated by
the behavioural models from 2013 to 2015 ranged from 0.013 to 0.045 mg l$^{-1}$; hence the width
of the credibility interval was 110% of the median (0.028 mg l$^{-1}$). The large uncertainty in the
calibration data, along with a lack of long-term information, also prevents including more
detailed processes in the soil routine.
To reduce uncertainty in prediction and to build more complex models, several options exist
to improve information content in the data. As stated by Jackson-Blake et al. (2015b), "the
key to obtaining a realistic model simulation is ensuring that the natural variability in water
chemistry is well represented by the monitoring data". The monitoring strategy adopted in the
Kervidy-Naizin catchment should theoretically enable to capture the natural variability in
stream SRP concentration, because sampling took place during two contrasting water years,
during different seasons and at a high frequency during 14 storm events. The analysis of
uncertainty in the data shows that a large part of uncertainty in "observed" SRP concentration
originates from sample storage, both unfiltered between the time of autosampling and manual
filtration and between filtration and analysis. This is due to SRP being non-conservative.
Thus, there is room for improvement in reducing storage time, without increasing further the
monitoring frequency. In this respect, the primary interest of investing in high frequency





bankside analysers would lie in their ability to analyse water samples immediately in addition to providing near continuous data. Because bankside analysers perform measurements in relatively homogeneous conditions, unlike the manual and autosampler data for which storage time of filtered and unfiltered samples vary, a finer quantification of uncertainty in the measurement data would be possible (e.g. Lloyd et al., 2015).

## 5    Conclusion

The TNT2-P model was capable of capturing daily variation of SRP loads, thus confirming the dominant processes identified in previous analyses of observation data in the Kervidy-Naizin catchment. The role of hydrology in controlling connectivity between soils and streams, and the role of soil Olsen P, soil moisture and temperature in controlling SRP solubility have been confirmed. The lack of any representation of the short-term effect of management practices did not seem to penalize the model's performance. Their long-term effect on the soil Olsen P could be simulated with an independent model or through an additional sub-model if a longer period of data was available to calibrate it. The modelling approach presented in this paper included an assessment of the information content in the data, and propagation of uncertainty in the model's prediction. The information content of the data was sufficient to explore dominant processes, but the relatively large uncertainty in SRP concentrations would seemingly limit the possibility for including more detailed processes into the model. Data from near continuous bankside analyser will probably allow calibrating more detailed models in the near future.

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

**Acknowledgements**
This work was funded by the ''Agence de l'Eau Loire Bretagne'' via the ''Trans-P project''.
Long-term monitoring in the Kervidy-Naizin catchment is supported by ''ORE AgrHyS''.
Data of ''ORE AgrHyS'' can be downloaded from http://www6.inra.fr/ore_agrhys/Donnees.





Table 1: Initial parameter ranges in the hydrological and soil phosphorus sub models.

| | Abbreviation | Unit | Hydrological (H), Phosphorus model (P) | Range poorly drained soils (min-max) | Range well drained soils (min-max) |
|---|---|---|---|---|---|
| **Lateral transmissivity at saturation** | T | $m^2\ d^{-1}$ | H | 4-8 | -> x1.5 |
| **Exponential decay rate of hydraulic conductivity with depth** | m | $m^2\ d^{-1}$ | H | 0.02-0.2 | 0.02-0.2 |
| **Soil depth** | ho | m | H | 0.3-0.8 | -> x1 |
| **Drainage porosity of soil** | po | $cm^3\ cm^{-3}$ | H | 0.1-0.4 | -> x1 |
| **Regolith layer thickness** | h1 | m | H | 5-10 | -> x4 |
| **Exponent for evaporation limit** | α | - | H | 8 (fixed) | -> x1 |
| **kRC parameter for capillary rise** | kRC | - | H | 0.001 (fixed) | -> x1 |
| **n parameter for capillarity rise** | n | - | H | 2.5 (fixed) | -> x1 |
| **Drainage porosity of regolith layer** | p1 | $cm^3\ cm^{-3}$ | H | 0.01-0.05 | -> x1 |
| **Background P release coefficient for subsurface flow** | $Coef_{SRP\ overland}$ | - | P | 0-0.015 | -> x1 |
| **Background P release coefficient for overland flow** | $Coef_{SRP\ sub-surface}$ | - | P | 0-0.25 | -> x1 |
| **Temperature coefficient 1** | T1 | - | P | 5-10 | -> x1 |
| **Temperature coefficient 2** | T2 | - | P | 2-10 | -> x1 |





| Soil moisture coefficient | S1 | - | P | 0-2 | -> x1 |
|---|---|---|---|---|---|
| **SRP concentration in deep flow** | SRP_deep | mg l$^{-1}$ | P | 0-0.007 | -> x1 |

Table 2: Starting and ending dates of periods studied

| Name | Starting date | Ending date |
|---|---|---|
| **Autumn 2013** | 01 October 2013 | 31 December 2013 |
| **Winter 2014** | 01 January 2014 | 31 March 2014 |
| **Spring 2014** | 01 April 2014 | 31 July 2014 |
| **Autumn 2014** | 01 October 2014 | 31 December 2014 |
| **Winter 2015** | 01 January 2015 | 31 March 2015 |
| **Spring 2015** | 01 April 2015 | 31 July 2015 |

Table 3: Sensitivity analysis of the model to 18 model parameters (insignificant ., important *,
critical ***). Parameters significations are detailed in Table 1.

| | discharge | baseflow SRP load | storm SRP load |
|---|---|---|---|
| **T (poorly drained soils)** | . | *** | *** |
| **m (poorly drained soils)** | *** | *** | *** |
| **ho (poorly drained soils)** | *** | *** | . |
| **po (poorly drained soils)** | *** | *** | *** |
| **h1 (poorly drained soils)** | *** | . | . |
| **p1 (poorly drained** | *** | *** | * |



| | | | |
|---|---|---|---|
| soils) | | | |
| T (well drained soils) | . | *** | *** |
| m (well drained soils) | *** | *** | *** |
| ho (well drained soils) | *** | *** | . |
| po (well drained soils) | *** | *** | *** |
| h1 (well drained soils) | *** | *** | . |
| p1 (well drained soils) | *** | *** | * |
| Coef_sub-surface | . | *** | . |
| Coef_overland | . | *** | *** |
| SRP_deep | . | . | * |
| S1 | . | *** | *** |
| T1 | . | *** | *** |
| T2 | . | *** | *** |



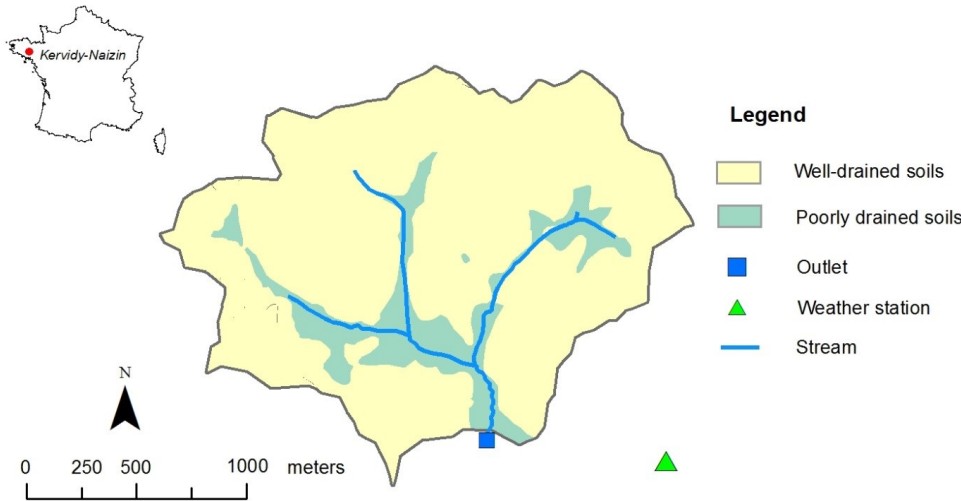

2    Fig. 1. Soil drainage classes in the Kervidy-Naizin catchment, Curmi et al. (1998)

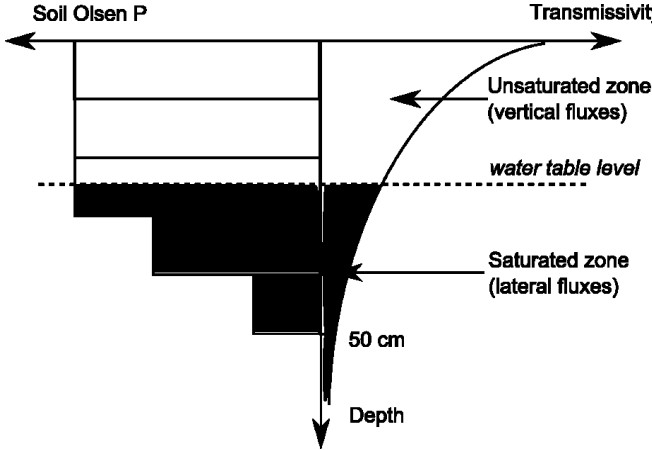

4    Fig. 2. Description of soil hydraulic properties and phosphorus content with depth





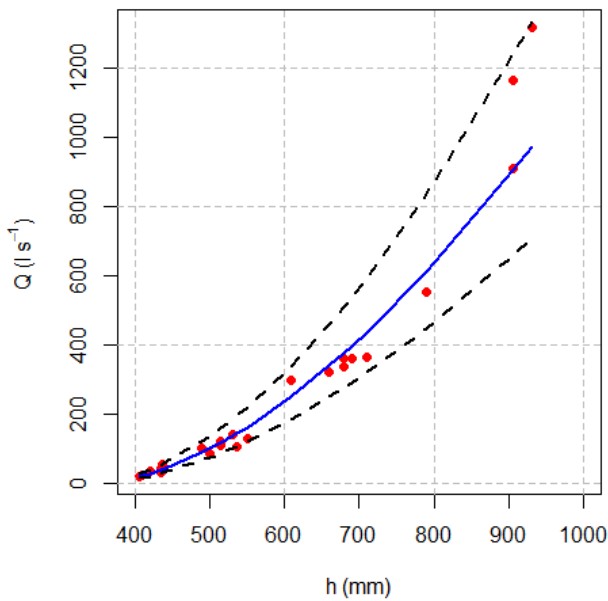

Fig. 3 : Rating curve in Kervidy-Naizin; acceptability bounds derived from 90% prediction
interval (blue line: fitting regression; black dots: 90% prediction interval). Red dots represent
the original discharge measurements used to calibrate the stage-discharge rating curve
(Carluer, 1998).

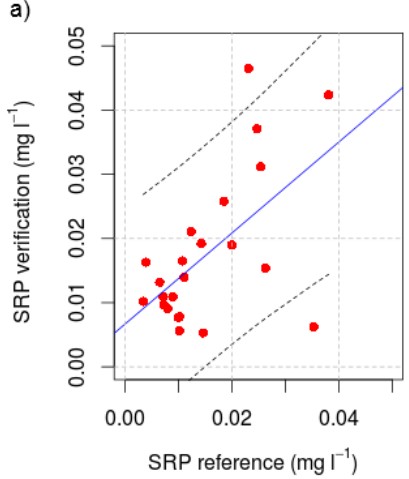
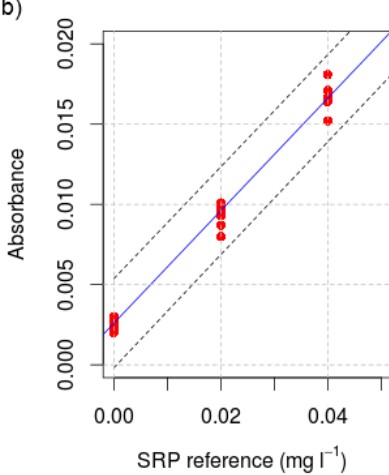




Fig. 4: a) linear regression model linking the reference data and a verification dataset; b)
measurement error as estimated from a repeatability test performed by the lab in charge of
producing reference data (blue line: fitting regression; black dots: 90% prediction interval).

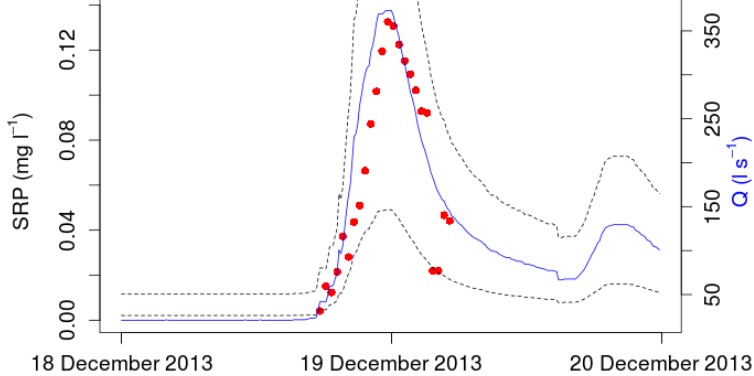

Fig. 5: Example of an empirical concentration – discharge model; acceptability bounds
derived from 90% prediction interval. Red circles represent the SRP measurements.

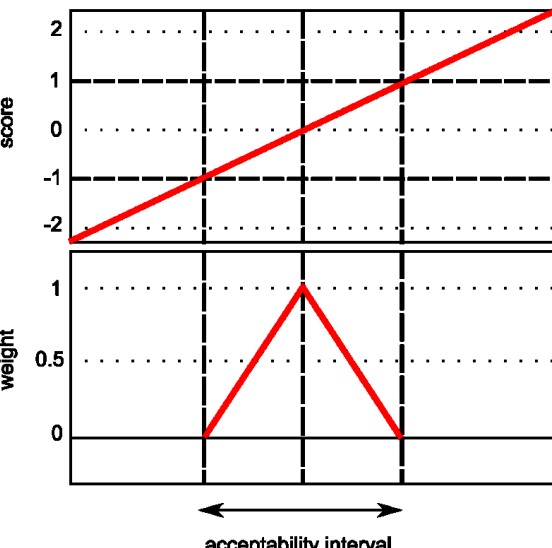

Fig. 6 : a) normalized scores; b) triangular weighting function



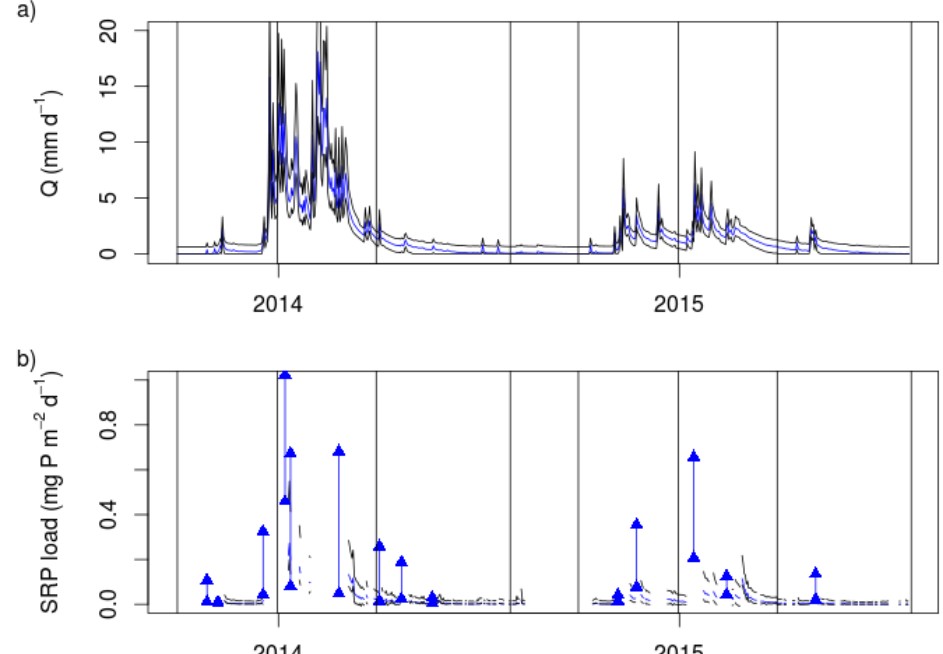

Fig. 7: Acceptability limits for daily discharge (a) and SRP load (b). Blue lines represent best
estimates; black lines represent the acceptability limits. Storm loads acceptability limits are
represented by vertical blue lines. Black vertical lines represent the starting and ending dates
for each season (table 2).





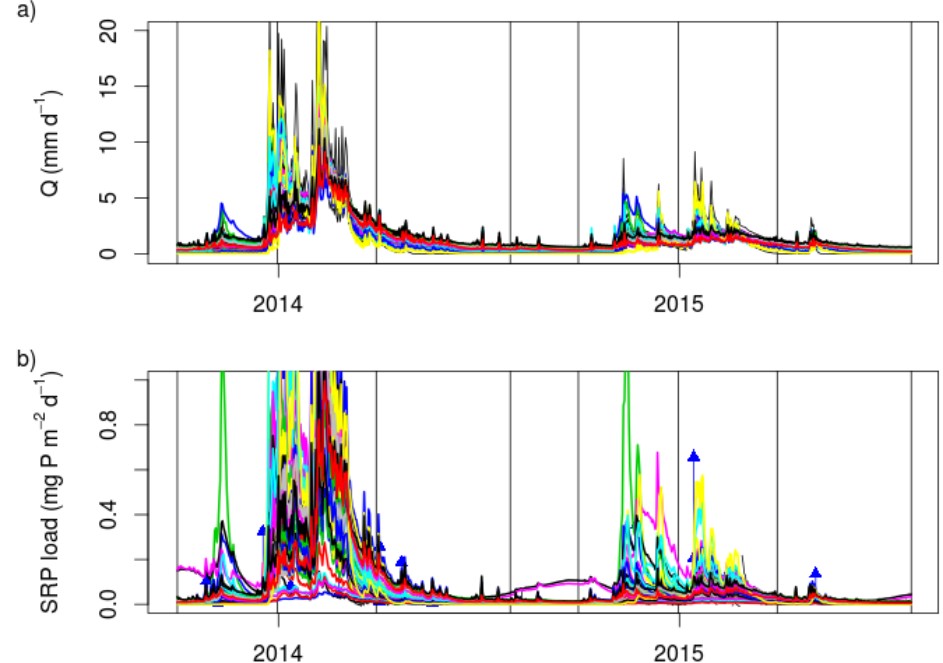

2    Fig. 8: Example of 50 model runs simulating discharge (a) and daily load (b). Vertical lines

3    represent the starting and ending dates for each season (table 2).





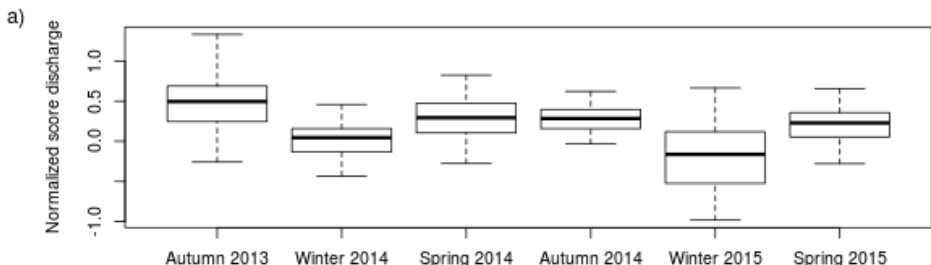

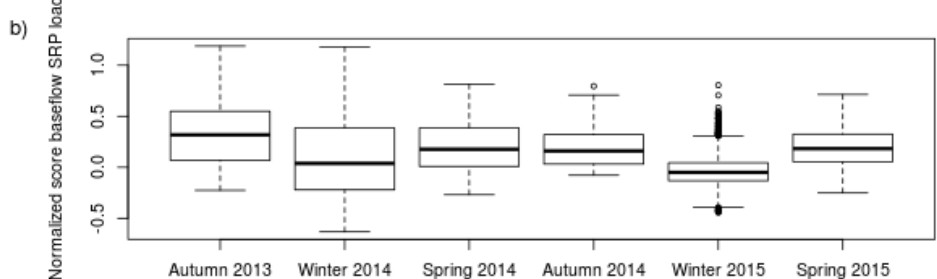

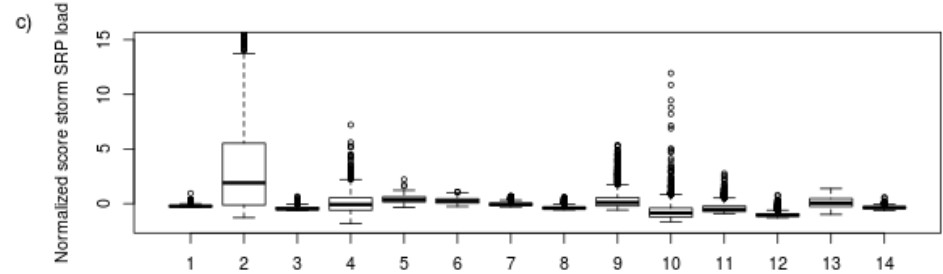

2  Fig. 9: Normalized score for daily discharge (a), baseflow SRP load (b) and storm SRP load

3  (c).





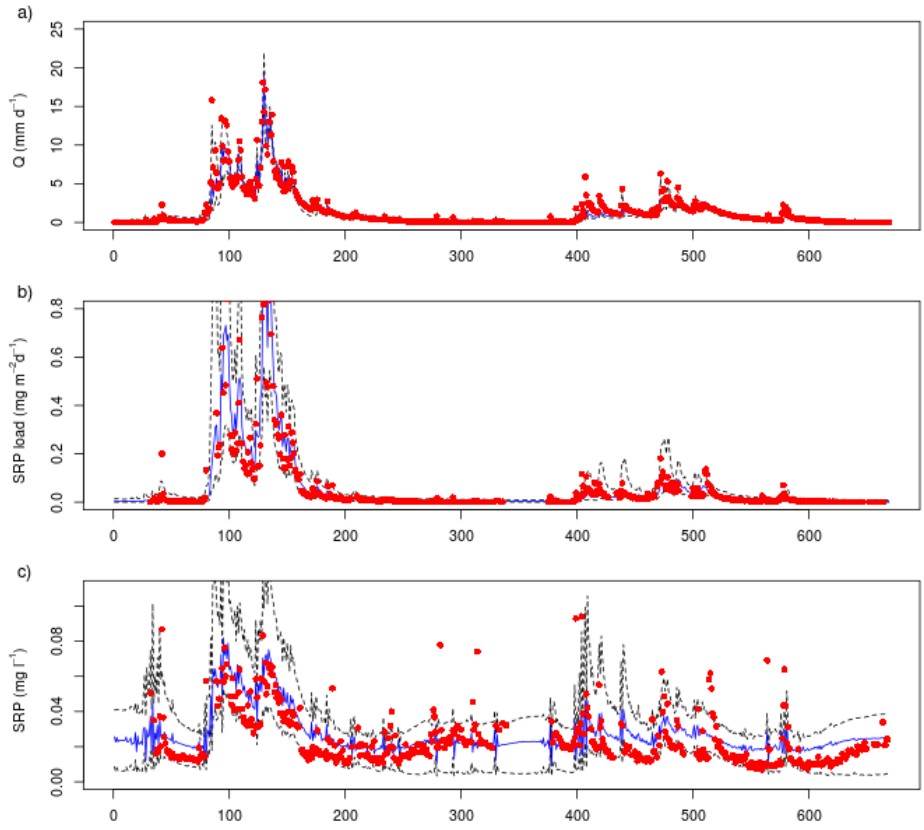

2 Fig. 10: Median and 95% credibility interval for daily discharge (a), SRP load (b) and SRP

3 concentration (c). Red circles represent observational data.

