# Peer review of "Uncertainty assessment of a dominant-process catchment"

_Hydrology and Earth System Sciences, 2015_

## Referee Comment (RC1) · T. Krueger (Referee) · 4 Feb 2016

I enjoyed reading this paper. It is structured and clear and makes useful contributions to the GLUE limits of acceptability (LOA) approach and model evaluation in a water quality context more widely. The authors took great care to derive the LOA from independent evidence including a small number of repeat samples (which are hard to come by). I think the GLUE LOA approach shows its strength when the LOA are empirically grounded and not defined ad hoc as in some of the earlier 'proof of concept' applications. Hence this study will help greatly to operationalise the method. It does help that the authors come up with a model whose complexity is attuned to the data availability. This means the model actually manages to operate within the LOA without the authors having to unduly relax them post hoc. Where they relax the LOA this is done sensibly and well justified. Here again the paper goes beyond earlier GLUE LOA

studies (including my own) where the relaxation of LOA was a bit unsatisfactory.

One methodological inconsistency, however, is the translation of probability density functions (from regressions) into LOA and triangular weighting functions (sections 2.3.1-2.3.2, P11-13). Not only does translating pdfs into LOA by using the central 90% confidence bounds seem arbitrary. What is more, arguably, if you are happy with the regression (and the assumptions that come with it) then you have a probabilistic representation of uncertainty that you can use in GLUE. So why not use this information? If you're not happy with the regression (I wouldn't most of the time) then why not use a method more attuned to the assumptions you're happy to make? Examples in the GLUE LOA context are: Pappenberger et al. (2006), Krueger et al. (2010, 2012), Westerberg et al. (2011). I think this inconsistency weakens the method considerably when it is exactly the empirical grounding of the LOA and weighting functions that make it so valuable, as I argue above. Preferably, the method would be changed accordingly – or convincing arguments for the chosen approach presented in the paper.

Another aspect that is not entirely clear is the reasoning behind the 2-day aggregation of SRP loads (P12, L29-P13, L7). I see that you compare the model to total storm loads that way. But why not compare the model to daily loads (which it can simulate)? This would be a much stronger test of the model. Arguably, why need a daily resolution model at all if you are only interested in storm loads? And if you need the processes that the daily resolution covers, why not test these with data at the same resolution (which you have)? Preferably, the study would be adapted to take this into account – or we need convincing arguments in the paper why the present approach was chosen despite the concerns identified.

I'm also missing all the earlier GLUE LOA applications in the field of water quality modelling. Reviewing these in the introduction and discussing your own results against what they had to say would increase the scholarly impact of the paper I would say. Among those are: Page et al. (2003, 2004), Rankinen et al. (2006), Quinton et al. (2011), Krueger et al. (2012). Even if some of these don't call their approach LOA, they

nevertheless, through the use of fuzzy performance measures with a finite support and multiplicative aggregation, effectively apply LOA within GLUE!

Specific points

P2, L5-7: It would be good to also refer to ecological impacts here.

P2, L13-15: The P fractionation in transit (e.g. resorption) would be important, too.

P3, L31-P4, L1: Here you could cite Krueger et al. (2012) where we dealt with evaluation data uncertainty (suspended solids, TP) explicitly in a GLUE limits of acceptability framework, albeit with even simpler models at finer time scales.

P4, L2-5: The grab sample uncertainty discussion could be usefully enhanced by referencing McMillan et al. (2012) where we discuss these issues at length by synthesising a large body of work.

P4, L3-5: Grab samples also represent a snapshot at a given point in the stream (e.g. Rode & Suhr, 2007).

P5, L12 and elsewhere: Please specify what +/- represents – one standard deviation?

P7, L13f: I think there should be "model" at the end of the title.

P10, L4-17: Here or elsewhere it would be good to note that no long-term depletion of soil P pools was modelled, i.e. effectively assuming steady state. This would also be an interesting point for discussion.

P11, L4-16: Here especially it would be good to cite other GLUE LOA studies in water quality modelling, see above.

P12, L20-22: Does this imply that no intercept was fitted in the regression equation? Would be good to clarify either choice.

P14, L7-9: Why were the weights summed (average) and not multiplied in keeping with the LOA concept? Krueger et al. (2012) discuss this.

P16, L23-25: No. What you must say is that you cannot reject this set of processes as a hypothesis of dominant control given the available evidence! There is no confirmation here, only a failed rejection.

Section 4.2, P17-19: Here I'm missing a discussion of the neglect of farm management practices in the model – which are vitally important if the model is to eventually have any bearing on catchment management.

P18, L5-8: If you want to make this point then you should also discuss what benefit the finer resolution of the SW-GW interactions brings given that the subsequent P processes are much coarser (e.g. there are no hyporheic zone P reactions).

Fig. 7: The 1st storm in (a) is not easy to see (lines too close together) – consider different x-scale or 2 panels or else. In (b) the mix of lines and vertical lines with triangles to represent the LOA at the different resolutions (storms vs. baseflow) is confusing. Best would be to evaluate the model during the storms at the same resolution as during baseflow (daily, see above). If you can convince that this is not necessary then think of a different representation, maybe only the vertical lines but making the triangles smaller.

Fig. 8: Can't distinguish the lines and the LOA to get a sense of model fit. Consider scaling x differently, see above, and making model lines smaller and in same colour. Emphasise LOA lines (perhaps move in front of model lines).

References

Krueger, T., J. Freer, J. N. Quinton, C. J. A. Macleod, G. S. Bilotta, R. E. Brazier, P. Butler, and P. M. Haygarth (2010), Ensemble evaluation of hydrological model hypotheses, Water Resour. Res., 46, W07516, doi: 07510.01029/02009WR007845.

Krueger T, Quinton JN, Freer J, Macleod CJA, Bilotta GS, Brazier RE, Hawkins JMB, Haygarth PM. 2012. Comparing empirical models for sediment and phosphorus transfer fromsoils to water at field and catchment scale under data uncertainty. European

Journal of Soil Science 63(2): 211–223.

Page, T., K. J. Beven, J. Freer, and A. Jenkins (2003), Investigating the uncertainty in predicting responses to atmospheric deposition using the model of acidification of groundwater in catchments (MAGIC) within a generalised likelihood uncertainty estimation (GLUE) framework, Water Air Soil Pollut., 142(1-4), 71-94.

Page, T., K. J. Beven, and D. Whyatt (2004), Predictive capability in estimating changes in water quality: Long-term responses to atmospheric deposition, Water Air Soil Pollut., 151(1-4), 215-244.

Pappenberger, F., P. Matgen, K. J. Beven, J. B. Henry, L. Pfister, and P. Fraipont de (2006), Influence of uncertain boundary conditions and model structure on flood inundation predictions, Adv. Water Resour., 29(10), 1430-1449.

Quinton, J. N., T. Krueger, J. Freer, G. S. Bilotta, and R. E. Brazier (2011), A case study of uncertainty: Applying GLUE to EUROSEM, in Handbook of Erosion Modelling, edited by R. P. C. Morgan and M. A. Nearing, pp. 80-97, Blackwell Publishing Ltd, Chichester.

Rankinen, K., T. Karvonen, and D. Butterfield (2006), An application of the GLUE methodology for estimating the parameters of the INCA-N model, Sci. Total Environ., 365(1-3), 123-139.

Rode M, Suhr U. 2007. Uncertainties in selected river water quality data. Hydrology and Earth System Sciences 11(2): 863–874.

Westerberg, I., Guerrero, J.-L., Seibert, J., Beven, K. J., and Halldin, S.: Stage-discharge uncertainty derived with a non-stationary rating curve in the Choluteca River, Honduras, Hydrol. Process., 25, 603–613, doi:10.1002/hyp.7848, 2011.

---

## Referee Comment (RC2) · P. G. Whitehead (Referee) · 9 Mar 2016

This is an excellent paper, addressing the major issue in water quality modelling of how to minimise the complexity of any model but at the same time capture the key dynamics and processes operating. The authors use a upstream agricultural catchment which is fairly small and thereby they minimise the problems of trying to model the instream processes, as well as the terrestrial and soil systems. So this simplifies the modelling but one question is 1. Are there any instream dynamics occurring such as precipitation of P onto the sediment bed and remobilisation during storm events? The paper of Wade et al referred to the in paper addresses some of these issues.

The model of the soils system is kept simple and makes use of the standard Olsen P soil measurements to keep track of P loading on the soils.

[Figure]

What is really excellent about this paper is the combination of the statistical and sensitivity techniques utilised. The combination of the GLUE methodology but framed within a realistic limits of acceptability approach is a very good strategy, but this is further enhanced by the use of the Hornberger-Spear GSA methodology, to investigate parameter uncertainty. Whilst this proved to be a very useful strategy to evaluate the parameters, the work could have been taken a stage further to utilise the GSA technique to evaluate the distribution of parameters and the Kolmogorov–Smirnov Statistics for distribution separation. This tells you which parameters are controlling the dynamic behaviour and ranks the parameters. This is described in the 2 papers below, and can be used as a means of improving the model fit, by focusing on the calibration of the most sensitive parameters. Whilst I do not think this analysis is needed for this paper, it would make an interesting follow on piece of work.

WHITEHEAD, P.G., HORNBERGER, G.E. (1984), Modelling algal behaviour in the River Thames, Water Research, Vol.18, pp. 945-953.

WADE, A.J. WHITEHEAD, P.G., HORNBERGER, G.M., SNOOK, D. (2002) On Modelling the flow controls on macrophytes and epiphyte dynamics in a lowland permeable catchment: the River Kennet, southern England. Sci Tot Environ. Vol 282-283 pp. 395-417.

---

## Author Comment (AC1) · 24 Apr 2016

Dear Editor and Referees,

We appreciated the constructive criticisms of the two referees Tobias Krüger and Paul Whitehead. We have addressed each of their concerns as outlined below.

RC1

One methodological inconsistency, however, is the translation of probability density functions (from regressions) into LOA and triangular weighting functions (sections 2.3.1-2.3.2, P11-13). [. . .]

1. We agree that a triangular measure might not always be the best approximation to the normal distribution. Hence we will reweight the contributions using the statistical

deviation weights (truncated to 90 %).

Another aspect that is not entirely clear is the reasoning behind the 2-day aggregation of SRP loads (P12, L29-P13, L7) [. . .]

2. I understand the concern: when aggregating the data at a coarser resolution than that of the simulation, constrains on the model become loose and there is a risk of accepting models that would not have been accepted otherwise. Here is the reason for two-day aggregation: in the study catchment, storm events usually affect SRP concentrations during around 24h. A rainfall at the beginning of the day will lead to high load during the first day and "almost baseflow load" during the second day. If the rainfall occurs at the end of the day, then the high load will occur during the next day and the mean load of the first day will be "almost baseflow load". With the daily information we use in this modelling study, the model does not know what time the rainfall occurred hence the 2-day aggregation, which represents the load during the storm event (plus some "baseflow load" before or after the storm event).

We will amend the manuscript: "A 2-day aggregation was necessary here because increased SRP load as a response to each storm event could occur either mainly during the day of the rainfall (if the rainfall occurred early in the morning) or mainly during the day following the rainfall (if the rainfall occurred late in the evening), and with the daily resolution of the input data and model simulation, the information about the timing of the rainfall event was not available to the model."

I'm also missing all the earlier GLUE LOA applications in the field of water quality modelling [. . .]

3. Some of the references suggested will be added.

Specific points

P2, L5-7: It would be good to also refer to ecological impacts here.

4. The manuscript will be amended as suggested "serious hazard to ecosystems and

humans"

P2, L13-15: The P fractionation in transit (e.g. resorption) would be important, too.

5. The manuscript will be amended as suggested "as well as the potential P resorption during transit"

P3, L31-P4, L1: Here you could cite Krueger et al. (2012) where we dealt with evaluation data uncertainty (suspended solids, TP) explicitly in a GLUE limits of acceptability framework, albeit with even simpler models at finer time scales.

6. This reference will be added where suggested.

P4, L2-5: The grab sample uncertainty discussion could be usefully enhanced by referencing McMillan et al. (2012) where we discuss these issues at length by synthesizing a large body of work. P4, L3-5: Grab samples also represent a snapshot at a given point in the stream (e.g.Rode & Suhr, 2007).

7. The manuscript will be amended as suggested. "Grab sample data represent a snapshot of the concentration at a given time of the day, which can differ from the flow weighted mean daily concentration (McMillan et al. 2012), and a specific point in the stream cross-section, which can differ from the cross section mean concentration (Rode and Suhr, 2007)."

P5, L12 and elsewhere: Please specify what +/- represents – one standard deviation?

8. We will add "$\pm$ stand deviations".

P7, L13f: I think there should be "model" at the end of the title.

9. "model" will be added at the end of the title.

P10, L4-17: Here or elsewhere it would be good to note that no long-term depletion of soil P pools was modelled, i.e. effectively assuming steady state. This would also be an interesting point for discussion.

10. The manuscript will be amended as suggested in the "Methods" section. "No long-term depletion of the different P pools was modelled, because P export from the catchment was small compared to the size of soil and sub-soil P pools." The fact that long term evolution of soil P content was not modelled by TNT2-P (and our recommendation not to try and couple TNT2-P with a long term soil P content model but instead generate new map with a long term model and use these maps as input data to TNT2-P) is already part of the discussion.

P11, L4-16: Here especially it would be good to cite other GLUE LOA studies in water quality modelling, see above.

11. Here we are only citing general papers (including early papers) about the concepts and the philosophy of GLUE. References about applications to different field (including water quality) will be cited earlier in the manuscript (see comment 3).

P12, L20-22: Does this imply that no intercept was fitted in the regression equation? Would be good to clarify either choice.

12. There is an intercept. The fact that we assumed no bias due to sampling time does not mean that other sources of variability between the two datasets could not create a bias (for example the different storage time). To clarify this point, we will add the equation of the regression "y=a *x +b" P14, L7-9: Why were the weights summed (average) and not multiplied in keeping with the LOA concept? Krueger et al. (2012) discuss this.

13. I agree and I will multiply the weights in the revised manuscript.

P16, L23-25: No. What you must say is that you cannot reject this set of processes as a hypothesis of dominant control given the available evidence! There is no confirmation here, only a failed rejection.

14. I agree. The manuscript will be amended as suggested "The fairly good performance of TNT2-P at simulating SRP loads provides further support that the hydrological and biogeochemical processes included into the model are dominant controlling factors in the Kervidy-Naizin catchment (i.e. the modelling hypotheses could not be rejected based on this study)."

Section 4.2, P17-19: Here I'm missing a discussion of the neglect of farm management practices in the model – which are vitally important if the model is to eventually have any bearing on catchment management.

15. I disagree. The fact that the calendar of agricultural practices is not correlated with variability in stream SRP concentration is one of our starting hypotheses (section 2.1.3, page 6 line 25). Agricultural practices have an effect on long term change in soil P Olsen content. In the discussion paragraph 4.2, page 18 lines 18-25, we discuss whether a model to simulate long term change in soil P Olsen content should be coupled with TNT2-P or if such a model should be run separately.

P18, L5-8: If you want to make this point then you should also discuss what benefit the finer resolution of the SW-GW interactions brings given that the subsequent P processes are much coarser (e.g. there are no hyporheic zone P reactions).

16. I agree that some of the models mentioned here include routines to simulate stream P processing (including hyporheic processes). However our statement was not about surface water – groundwater interaction, it was about soil - groundwater interaction, which is represented in a more explicit way in TNT2-P compared to semi distributed models. To improve clarity, I will add "extend of the riparian wetland area".

Fig. 7: The 1st storm in (a) is not easy to see (lines too close together) – consider different x-scale or 2 panels or else. In (b) the mix of lines and vertical lines with triangles to represent the LOA at the different resolutions (storms vs. baseflow) is confusing. Best would be to evaluate the model during the storms at the same resolution as during baseflow (daily, see above). If you can convince that this is not necessary then think of a different representation, maybe only the vertical lines but making the triangles smaller.

17. About time resolution of model evaluation see response 2. The stream was dry before the first storm of the year (and after the storm) hence the line = 0. Because several people do not like the triangles, I will remove them.

Fig. 8: Can't distinguish the lines and the LOA to get a sense of model fit. Consider scaling x differently, see above, and making model lines smaller and in same colour. Emphasise LOA lines (perhaps move in front of model lines).

18. I agree it is difficult to read both the uncertainty intervals and the results of model runs in this figure. For this reason I will remove the uncertainty interval from the "model runs" figure (8) and place it next to the "acceptability intervals" figure (7). Please note: the "model runs" figure shows only 50 runs, hence this illustration does not aim to be used for counting how many models fit. You can get a better sense model fit with figure 9 and with the number of models fulfilling the selection criteria as described in the text.

---

## Author Comment (AC2) · 24 Apr 2016

Dear Editor and Referees,

We appreciated the constructive criticisms of the two referees Tobias Krüger and Paul Whitehead. We have addressed each of their concerns as outlined below.

RC2

[. . .] Are there any instream dynamics occurring such as precipitation of P onto the sediment bed and remobilisation during storm events? The paper of Wade et al referred to the in paper addresses some of these issues.

19. Our (unpublished) data suggest that resuspension of stream sediments would adsorb SRP during storm events, especially during the rising limb of the hydrograph.

[Figure]

In a previous study (Dupas et al. 2015 Hydrological Processes) resorption onto stream sediment during the rising limb of the hydrograph was hypothesised as being a possible cause of the hysteresis loop in the C-Q relationships during storm events. We will add a sentence to mention this process "The reason for this simplification was that we lacked knowledge of SRP re-adsorption in downslope cells (or on suspended sediments in the stream network) and on the long-term fate of re-adsorbed SRP". But contrarily to Wade et al. we used our model in a small catchment to avoid having to simulate stream processes in order to focus on land-to-stream transfer.

The model of the soils system is kept simple and makes use of the standard Olsen P soil measurements to keep track of P loading on the soils.

What is really excellent about this paper is the combination of the statistical and sensitivity techniques utilised. The combination of the GLUE methodology but framed within a realistic limits of acceptability approach is a very good strategy, but this is further enhanced by the use of the Hornberger-Spear GSA methodology, to investigate parameter uncertainty. Whilst this proved to be a very useful strategy to evaluate the parameters, the work could have been taken a stage further to utilise the GSA technique to evaluate the distribution of parameters and the Kolmogorov–Smirnov Statistics for distribution separation. This tells you which parameters are controlling the dynamic behaviour and ranks the parameters. This is described in the 2 papers below, and can be used as a means of improving the model fit, by focusing on the calibration of the most sensitive parameters. Whilst I do not think this analysis is needed for this paper, it would make an interesting follow on piece of work.

20. I agree and I would like to point the fact that the GLUE analysis for the hydrological model relies on a previous sensitivity analysis which identified the most sensitive hydrological parameters. The manuscript will be amended as suggested to include this suggestion as a perspective "This identification of sensitive parameters can be used in future application of the TNT2-P model in the study catchment, as suggested by Whitehead and Hornberger (1984) and Wade et al. (2002b)."

With best regards,

On behalf of all co-authors,

Rémi Dupas

---

## Author Response (AR1)

**Editor Decision: Publish subject to minor revisions (Editor review)** (03 Jun 2016) by Dr. Jim Freer
Comments to the Author:

The two reviewers agree the paper is a good contribution to the field of uncertainty evaluation of water quality models, the paper is on the whole well presented and I agree is fit for publication into HESS. The reviewers have clearly taken some time to read the paper thoroughly and their expertise on this matter is appreciated and equally where necessary the authors have responded well. I can see they will address all the points made by the reviewers in a satisfactory manner. However I have a few additional comments to make on the paper that I think need to be improved as well as the reviewer comments, these are generally minor but I think will improve the paper. However one is major and I think critical to address and was surprised this was not brought up in the review process (or I have miss-understood what has happened in the paper):

My major comments are on the expressions of the uncertainty limits which are for this paper very large in both cases (flow and SRP). The question I put to the authors is how these errors can be justified (no comment is made to the extent of these in the paper and it is absolutely critical to the whole model evaluation conclusions drawn). First the authors have chosen a parametric regression approach to the uncertainties in discharge – so the first point is are the assumptions in this approach valid (perhaps they could relate to Coxon et al. 2015) which used a non-parametric approach. 39% errors need to be justified I believe.

Secondly and more critically what the authors have done to calculate an 'observational uncertainty' value for the storm event behaviour is to use effectively a model error approach to their found discharge-concentration relationship (and perhaps pulled together from multiple events but not 100% clear and could do with some figures to better identify their approach). That is not daily observational error representation from the data they have. That is the error associated with using a simple expected relationship between discharge and concentration that will inevitably lead to much wider uncertainty limits than would be expected for a daily mean flow uncertainty value in my view. Furthermore they do not show how much this function is being used to extrapolate these errors elsewhere and if that can be justified to how these were lumped or otherwise in the first place.

Surely a much more sensible approach to trying to characterise the mean error of SRP for a day when they have event information is to first derive the expected uncertainty in the measurements themselves (which the authors have) and then resample these for these events to gain the actual observational uncertainty mean daily limits. Then they would be able 'in extrapolation' to take the discharge-concentration mean value and apply such limits to these. I would argue where there is data available to assess the actual mean daily uncertainty from high resolution samples then an appropriate method would show the range of uncertainty would be very different to those generated by the authors in this manuscript. I suggest this must be better evaluated before the paper is fit for publication. I further add their is no discussion of the chosen approach and any implications nor any comment as to how high the ranges of error are for the LoA compared to the general range of concentrations in the time series.

We extended the paragraph on discharge uncertainty to explain why it appeared to be so high:

"This uncertainty interval is in the higher range of values found in other studies, e.g. Coxon et al. (2015) who found that mean discharge uncertainty was generally between 20% and 40% in 500 catchments of the United Kingdom. This relatively large uncertainty interval is due to the fact that it was derived from a prediction interval rather than a confidence interval (the 90% confidence interval of the log-log linear regression would be 14% of the mean discharge value during the study period). This choice of a relatively large acceptability interval counterbalances the fact than other sources of uncertainty (e.g. uncertainty in rainfall) were not accounted for in the discharge limits of acceptability. Moreover, the high percentage often represents a low absolute value because daily discharge was below 2 mm d-1 during 78% of the time during the study period." Page 12 line 3-13.

About estimation of SRP load during storm events: a different empirical model was fit for each event separately and the models were not applied to multiple events. When a storm event was not monitored at high frequency, the model was not evaluated for this storm event. We amended the manuscript:

"An empirical model was used to fit to each storm event monitored separately" page 13 line 16.

And

"During days with a storm event not monitored at high frequency with an autosampler, we considered that the grab sample data did not contain enough information to derive an acceptability interval for daily SRP load; hence simulated load was not evaluated for events not monitored at high frequency." Page 14 lines 3-6.

Similar to discharge, we added a sentence to comment on the fact that the large concentration uncertainty (in %) is actually small in absolute value :

"As for discharge estimates, the high percentage represents a small absolute value (0.03 mg l-1) during baseflow periods." Page 13 line 11-13.

We disagree that the method suggested here is better than ours:

First because observational concentration uncertainty for autosampler data is expected to be higher than just analytical uncertainty (because the samples are not filtered and analysed immediately when collected with an autosampler). A different method to assess uncertainty is used here to extend the acceptability interval with a 1 – 1.6 ratio (see manuscript).

Second because the empirical model is necessary to assess daily (or 2-daily) mean SRP concentration during storm events because autosampler were not running from 00:00 am to 00:00 pm during days with a storm event. The data points collected are rather biased towards the storm events itself (usually during 12h), therefore extrapolation is needed to assess the mean daily (or 2-daily) concentration. This can be clearly seen from the storm event given as an example in Figure 4 or in the other storm events shown in the supplementary material.

My other minor points include:

1) The intro states 'In this paper we strive to identify and quantify the different sources of uncertainty in the data when the required quality check tests have been performed' – yes but it should be made clear there are only some of the observational uncertainties that are dealt with here, and indeed maybe not some of the main uncertainties….

Response 2:

We agree. Other sources of uncertainty are mentioned in the Materials and methods.

"Input data, such as weather and soil Olsen P data, also contained uncertainty which were not accounted for explicitly in the limits of acceptability due to a lack of data to quantifying them." Page 11 lines 23-25.

See also response 1 about the large uncertainty interval.

2) Whilst one approach to understanding observational uncertainties in water quality data is to compare when samples are taken this is not a full characterisation of the potential errors.

Response 3:

We agree and we acknowledged this in the manuscript. The two samples taken during the same day during baseflow periods also had different storage time and independent lab analysis to account for many sources of variability:

"To assess uncertainty in daily SRP concentration related to sampling time, storage and measurement errors, a second grab sample was taken at a different time of the day (between 11:00 – 15:00 local time) in 36 instances during the study period. The second sample was analysed within 24h with the same method; this second dataset is referred to as verification dataset, as opposed to the reference dataset."

And indeed the estimated uncertainty is larger than that derived from a lab repeatability test:

"This method encompasses all various sources of uncertainty, which results in prediction intervals much wider than what would result from a mere repeatability test: at the median concentration (0.02 mg l-1), estimated prediction interval was 166% with this method versus 57% with a repeatability test (Fig. 4)."

3) The comments on page 8 about comparisons to TOPMODEL are to me confusing. It makes it sound like TOPMODEL did some form of explicit routing between 'grouped' hydrologically similar points, but it did not, and this is only available in Dynamic TOPMODEL (Beven and Freer, 2001).

Response 4:

We agree and we deleted this sentence on page 8.

We also amended the discussion:

"This could be achieved by grouping cells according to a hydrological similarity criterion like in  Dynamic Topmodel (Beven and Freer, 2001; Metcalfe et al., 2015) and do the same for similarity in soil P content." Page 19 line 13.

4) It makes no sense to me why this whole simulation is being run at 20m resolution. What is the point of this in terms of the landscape controls that need to be captured and the importance of such local parameterisation and interaction between cells that is either possible or critical. I do not see any justification in the spatial data presented nor the simple hypotheses presented about SRP that such fine detail is required. I feel the authors need to justify this far better in the paper (given they end up with only 2 drainage classes!)

Response 5:

The DEM resolution must be high compared to hillslope length for TNT2 (or TOPMODEL) to run correctly.

TNT2 is a fully distributed model, as explained in the materials and methods: "Based on these assumptions, TNT2 computes an explicit cell-to-cell routing of fluxes, using a D8 algorithm."

The two drainage classes determined values of hydrological parameters in the model but did not represent similar points grouped hydrologically.

5) Justify better how so many parameters can be really fixed and made homogeneous over the model domain please. No comments are made on this except the values are related to literature (does that mean they are all deterministic and not expected to vary in space?)

Response 6:

In the reference cited the initial parameter range was not derived from only one application of the model but rather from many of them in different contexts (but mainly in the same region). So it is a relatively large initial parameter range.

"Initial parameter ranges for the hydrological sub-model were based on literature-derived values from several previous studies in Western France (Moreau et al., 2013)" page 10 lines 1-3.

6) 15,000 simulations for 12 parameters is actually quite a small set. Please can the authors make comments about the acceptability of this sampling design given the needs of GLUE to sample the space effectively and how they confirmed this provided an acceptable simulation set.

Response 7:

We added a sentence to acknowledge this.

"The number of Monte Carlo realisations was constrained by the computation time required to run a spatially explicit model in this catchment." Page 14 line 9-12.

In the revised manuscript, this number was increased to 20,000 and results are similar to 15,000 runs.

7) On page 13 the authors state 'model runs must fall within the acceptability limits' – that would ONLY be the case if all errors in observations had been taken into account, but here as the authors make clear they are not including all sources of uncertainties so there is no need for this to be the case in their study.

Response 8:

We agree. See reponse 1 and response 2.

So given I have made perhaps the biggest critique of the paper on a point that I believe is fundamental to what has been evaluated I have put back the assessment to an editorial review for the improved manuscript, thanks, Jim Freer

[revised manuscript text omitted]

---

## Author Response (AR2)

Thanks for your response to the author's reviews and my own. But in fact I am not really yet in agreement with some of the rather quick justifications you have made to the manuscript on the basis of points I have raised so I would like these clarifying further please before I accept this for publication. These issues are important because if under the 'limits of acceptability' the methods are not clear how limits are quantified then in some sense they have no value as part of a considered experimental design.

Thank you for reading our manuscript one more time and for your suggestion to improve it. I realise that I had not understood properly some of the comments in your previous review, hopefully these responses will answer your questions.

Rémi Dupas

1) The authors haven't really justified either the use of using 'prediction intervals', nor that the error assumptions justify the parameteric approach chosen, on the basis of the observed information they have for the rating curve definition. I feel the authors need to deal with these matters better than they have done at the moment in formulation of GLUE LoA or discussion. Furthermore there is no clear rationale as to why a non-linear transformation in rainfall errors (not analysed) would in fact be a surrogate for using this parametric approach having wide uncertainties on the output as some kind of counter balance. I do not think that is well considered as written nor does it deal with the potential differences that might occur if the rainfall errors were . The same goes again for the discharge-concentration uncertainty. What is the proof the parameteric 'prediction interval' error model used relates to the observed error characteristics? – these are both important to get right and/or discuss the limitations/assumptions in them being used!

Response 1:

- In the previous version of the manuscript we justified the choice of a prediction interval rather than a confidence interval based on the fact that some sources of uncertainty were not included (rainfall, DEM, etc.) therefore we chose the largest interval among the two possible ones. This cannot be fully justified unless we analyse error in rainfall (and I do not have good data to do that) and in other sources of error (including those we have not thought about). Now we justify our choice more simply by saying that a prediction interval is an interval in which future observations will likely fall (whereas a confidence interval is an interval in which the mean of repeated observation will likely fall). Because in the TNT2-P model′s evaluation, we want observations to fall in the acceptability interval, a prediction interval is more appropriate.

  Lines 363-367: "A prediction interval is an interval in which future observations will likely fall, while a confidence interval is an interval in which the mean of repeated observation will likely fall. Because in the TNT2-P model′s evaluation, we want each observation to fall in the acceptability interval (section 2.3.3.), a prediction interval was more appropriate."

Of course this justification will only convince the reader if he is convinced that using statistical models was a good choice, which we justified as best as we can in the second point of this response.

- We have added a new discussion paragraph to discuss the drawbacks of using statistical models (three statistical models are used to derive acceptability limits: the rating curve, the SRP concentration uncertainty during baseflow period and the storm event interpolation model).

  Lines 661-671: "Finally, alternative methods to statistical models could be used to derive acceptability limits (in this study three statistical models are used: the rating curve, the SRP concentration uncertainty during baseflow periods and the storm event interpolation model) because statistical models have at least three shortcomings: i) they lump the uncertainty linked to the timing of sampling, the immediate or delayed filtration of the samples, the storage time and the analytical error; ii) the formula chosen adds error to the already existing measurement errors because empirical models are not perfect representation of the system dynamics; iii) they assume a parametric distribution and temporally independent errors which are not always verified in practice. As an alternative, non-parametric methods could be used, but these methods generally require a large number of data points and they are not suitable for extrapolation to extreme values."

- The last criticism in this comment concerns the "What is the proof the parameteric 'prediction interval' error model used relates to the observed error characteristics". A detailed response on the statistical C-Q model is given in comment 3, but we can already say here that we know that the analytical error is an underestimate of the true error in observation (which also includes delayed filtration and analysis) and that the statistical model adds some error related to the extrapolation.

2) I don't understand in the authors response what 'We disagree that the method suggested here is better than ours' is referring to. For a start I am not sure I stated a method was 'better', and secondly it is not at all clear what the context of this response is. So I would like that clarifying please. Perhaps it relates to 1) above…… but then it talks about discharge-concentration curves.

Response 2:

This was a response to the comment "Surely a much more sensible approach…" where it was suggested that we should consider analytical uncertainty rather than a C-Q model to assess uncertainty in SRP load during storm events (if I understood the comment).

The response was in two parts:

- The measurement uncertainty as assessed by the laboratory repetition test is an underestimate of the real uncertainty of autosampler data. The real uncertainty includes, in addition to analytical uncertainty, the issue of samples not immediately filtered and the effect of sample storage.

- We need a statistical model to extrapolate the concentration data from 12h of measurements to a 2-day mean concentration. This model will introduce more error (but this model´s error reflects the missing information originating from the fact that autosampler data did not cover the 2-day period which we use for evaluation).

We added the paragraph:

Lines 403-409: "Two reasons led us to use a statistical model (which also implies the assumption that errors are aleatory and temporally independent): i) the measurement uncertainty as assessed by the laboratory repetition test was an underestimate of the real uncertainty of autosampler data, because it does not include other major sources of error such as delayed filtration and sample decay during storage; ii) it was necessary to extrapolate the sub-daily observation to the daily resolution of the model. The limits of this choice will be discussed in section 4.3."

Concerning SRP concentration uncertainty during baseflow periods, analytical uncertainty is also an underestimate of the true uncertainty (because other sources of uncertainty such as timing of the grab sampling during the day, or sample storage also play a role), and this was the reason for the use of another statistical model. This was already explained in details in the manuscript.

As acknowledged and discussed in the discussion (see response 1) this choice has several limits which we believe will be solved in part with bankside analyser data, for which observation error will be easier to evaluate.

3) I'm sorry but I am not going to let this issue go of how you derive your load concentration uncertainties and at least make it clearer to the reader what you are doing because at the moment it does not seem consistent or it is certainly not written in a way that makes this year. To be clear from what I can read, you have constructed 'parametric prediction uncertainty limits' from the rating curve information. But then you actually do not use these in any way (as far as I can tell) to construct the load uncertainty estimates. You introduce a new model (and a very simple one at that), applied to every storm with a manually applied lag and you gain some very wide uncertainty bounds. Now there are good reasons why in that case the uncertainties will be large, and particularly if that very simple model is not good at describing the dynamics of the discharge-concentration dynamics. If fact as prediction uncertainties it could be argued it is significantly increasing what the potential error limits are in the observations of load. I understand that you need a 'model' (although I can still see other ways of doing this) because you wish to extrapolate beyond where you have ISCO samples over days. But that does not mean that you should attempt to be clear exactly what is being done, if that simple model is fit for purpose, the potential issues of increasing load uncertainty estimates over reasonable values if the model is not a description of the system and where you have data if you resampled the expected SRP uncertainties and the discharge uncertainties you have already calculated then what does that look like for the periods you can do this, that finally be clear that you do not seem to be using the uncertainties you have found in discharge to in any way quantify the prediction limits for this simple dischargeconcentration model but instead use standard statistical errors that are yet to be proven. To me this is currently not very clear and not necessarily consistent and it needs to be better explained and discussed……

Response 3:

- The reason why we used statistical models (one for the baseflow periods, one for the storm events) is explained in response 2, and we hope to convince the reader that it was a good choice considering the fact that analytical uncertainty is an underestimate of the true uncertainty and considering the need of extrapolation to the daily resolution of the model. The limits of this choice are now discussed in more details (see response 1).
- The method to derive load acceptability intervals from the 90% prediction interval of discharge and SRP concentration is given in the sentence: "The acceptability limit for daily load was estimated summing up relative uncertainty assessed for discharge and SRP concentration (in percentage)."
  We also had to "combine" the weights for discharge and SRP concentration, both being derived from the statistical model´s error distribution. The method to do this was missing in the manuscript, so added the information:
  Line 458-460: "To "combine" the weights derived from the rating curve and the SRP concentration statistical models, a kernel density estimate (with Gaussian smoothing kernel) was computed to fit 10,000 realisations of the multiplied error models."
- One last critic in this comment concerns the fact that if the C-Q models used to extrapolate SRP during storm events are bad models, the uncertainty interval will inevitably be large. This is true and the reader can make his own opinion on this by looking at the results for each individual model in the supplementary material. We have added a paragraph in the discussion to acknowledge this and to present the perspective that with a bankside analyser (running since April 2016 in this catchment) future work will not require such statistical models because near continuous data will be available and characterization of measurement error will be easier (no difference in the filtration protocol for grab samples and ISCO samples, no delay before analysis and constant analytical error).

Finally the acceptability intervals for storm event loads are also quite large because we stretched the intervals by a factor of 1 -1.6 based on the data we have which show that delayed filtration of autosampler data is causing an apparent loss of SRP.

Lines 424-428: "When comparing autosampler data with data from immediately filtered samples, the ratio obtained had the range 1-1.6 (mean = 1.3), hence autosampler data were underestimates of the true concentration, arguably through adsorption or biological consumption. We used the mean ratio to correct all storm uncertainty intervals by 30% and the range values to extend the upper limit by 60%. "

4) Regarding my minor point 1) noted previously the introduction still states 'In this paper we strive to identify and quantify the different sources of uncertainty in the data when the required quality check tests have been performed'. Again this needs to be clarified there what the limits of this is in the paper (so only discharge and the P data)

Response 4:

We have added this precision in the introduction: "discharge and SRP concentration data"

5) I do not see how the response to my minor point 5 on the application of homogeneous parameters across the domain has been answered in the response given.

Response 5:

Sorry I had misunderstood this comment.

For the hydrological parameters, we decided to use two soils classes according to the soil map of Curmi et al. (1998) because these authors have measured the hydraulic conductivity for 29 soil cubes in the two soil classes and they appeared to be different (see the following figure extracted from Curmi et al. 1998).

[Figure]

*Figure 7.* Saturated hydraulic conductivity of the well drained and poorly drained horizons.

We added the sentence:

Lines 383-387: "Experimental determination of saturated hydraulic conductivity (29 soil cores) by Curmi et al. (1998) showed significantly different values for soils classified as well-drained and poorly-drained in the Kervidy-Naizin catchment. The two units were treated as homogeneous, lacking information about the detailed variability in soil hydraulic characteristics at the model grid scale."

For the soil-P model, parameters were considered homogeneous because a previous study in the same catchment showed that the most important factor controlling SRP solubilisation in soils was P Olsen (see section 2.1.3 "Identification of dominant processes") therefore we concentrated our effort on producing a high resolution map of P Olsen (which is an input data to the model) but the parameters to relate this P Olsen to SRP concentration in the soil solution can be considered constant.

We added the sentence:

Line 301-306: "A previous study has shown that soil Olsen P was the most important factor controlling SRP solubilisation in soils of the Kervidy-Naizin catchment (see section 2.1.3.), so other parameters in the soil-P sub-model (section 2.2.2.) were treated as homogeneous in the catchment (the soil classification into well-drained and poorly-drained soils only concerned hydrological parameters)."

6) I think it needs to be justified far better than the response to minor point 6) is somehow justified for such a sparse sample. I'm not going to accept as a scientific evaluation that going from 15,000 – 20,000 simulations 'looked similar' without any justification of what that means. Nor that recognizes that one of the standpoints of using an approach such as GLUE is that the parameter space can be well sampled, or that if a sparse sample must be used there are experimental designs that improve the efficiency of sampling. In effect the authors have a parameter space they are trying to sample that even if they took 2 mid points on each axis this would require 2**30 simulations which is over 1 billion runs. So what convergence would be seen between 15-20K runs! Again the authors appear not to have recognized this at all and the response was not useful in my view and needs to be better justified if they are using GLUE.

Response 6:

We acknowledge 20,000 simulation is a low number and also the fact that the argument that going from 15,000 to 20,000 simulation gave similar results is more a qualitatively appreciation than a real scientific demonstration. We deleted the second part of the sentence (about the 15,000 to 20,000 test) but we maintained the first part where we state that the number of simulations was constrained by computation time.

Several techniques are proposed in the manuscript to solve this problem (some we applied and some we present as perspective):

- First, not all 30 parameters were varied, only 12, and this was already explained so we did not change the paragraph:

  Lines 320-325: "To reduce the number of model runs necessary to explore the parameter space using Monte Carlo simulations, several parameters were given fixed values, or a constant ratio between the two soil types was set (Table 1). In the hydrological sub-model, the parameters to vary were identified in a previous sensitivity analysis (Moreau et al., 2013). In the soil sub-model, all the parameters were varied. Finally, only 12 parameters were varied independently."

- As a perspective (and this was suggested by the reviewer Paul Whitehead), we suggest to use the result of our own sensitivity analysis to vary even less parameters in future applications of the model:

Line 463-465: "This identification of sensitive parameters can be used in future application of the TNT2-P model in the study catchment, as suggested by Whitehead and Hornberger (1984) and Wade et al. (2002b)."

- Also as a perspective we suggest a method to reduce computation time by introducing the concept of hydrological and chemical similarity. The following paragraph was extended to address this comment (additional sentences are underlined):

  Lines 593-603: "It would be interesting to test to what extent moving from an aggregative model with fully distributed information to a semi-distributed model would degrade the model performance while reducing computational cost. This could be achieved by grouping cells according to a hydrological similarity criterion like in Dynamic Topmodel (Beven and Freer, 2001b; Metcalfe et al., 2015) and do the same for similarity in soil P content. Reducing computation time is critical in the context of a GLUE analysis because this method requires the parameter space to be sampled adequately to identify those models to be considered acceptable. This is debatable here because 12 parameters were varied and only 20,000 model runs were performed. It is therefore possible that some regions of the parameter space with acceptable models might not have been sampled."

7) Similar issues of not really providing a useful response go with the response to minor point 4) and 5). First there is still seemingly no analyses of why 20m DEM resolution is needed that is explicitly written in the model setup, so if somehow the hillslope characterization is being lost if the resolution was lower then in what way is some critical threshold being reached for the D8 sharing downslope? How has that been confirmed given the simplifications in general in the model? I still don't see how this all squares with the authors own statement that the main SRP transportation processes are controlled hydrologically by valley bottom groundwater fluctuations (between page 6-7).

Response 7:

We have added the argument:

Lines 307-314: "A 20 m resolution was chosen for the DEM and the soil Olsen P raster map to allow a detailed representation of the interaction of the groundwater table (as simulated by the hydrological model) and the soil Olsen P (as given by the soil Olsen P map). Indeed the soil saturation and soil Olsen P can be very different in a narrow zone close to the stream compared to upslope due to the presence of a 5 to 50 m unfertilized buffer zone with lower Olsen P compared to fertilized fields. The Olsen P value close to the stream has a determining influence on SRP transfer, because this area is the most frequently connected to the stream, so a coarser resolution of the raster maps would degrade representation of the system."

Similarly to the criticism on the number of simulation and the number of soil hydrological classes, the only way to demonstrate that 20m resolution is really important would be to make a formal sensitivity analysis, which we did not do because i) we had already some expert knowledge on the best resolution (see the references about old applications of TOPMODEL in the catchment Bruneau et al., 1995; Franks et al, 1998 and all the TNT2 papers), the dominant processes to include, etc and ii) we were already constrained by calculation times to test all the different alternative possibilities.

Regarding minor point 5) here is nothing in the additional sentence added that at all discusses how these parameters are homogeneous across the catchment to the level they have been applied. No evidence is provided to say why that is realistic in the fully distributed model design or why 2 classes are the dominant hydrological-chemical classifications. This again needs to be improved and the responses were quite weak.

Response 8:

I have understood the criticism now and additional justification is given in response 5.

[revised manuscript text omitted]